# LoBCD-GW: A Fast and Data-Dependent Algorithm for Computing Gromov–Wasserstein Distance via Localized Block Coordinate Descent

**Jingni Song** [1] **Jiawei Huang** [1 2] **Kangke Cheng** [1] **Bangxian Han** [3] **Hu Ding** [1]

## Abstract

The Gromov-Wasserstein (GW) distance provides a powerful framework for aligning structured data by comparing the intrinsic geometries of metric measure spaces, and has become a fundamental tool in machine learning. Most existing methods leverage entropy regularization to reduce the computational complexity to $\mathbf{O}(n^3)$, where $n$ is the number of samples. However, this cubic time complexity remains a major bottleneck in large-scale applications, severely limiting the scalability. To address this challenge, we propose *LoBCD-GW*, an efficient GW optimization algorithm. Specifically, we reveal the data-dependent sparsity of large-magnitude updates to the coupling matrix and introduce a localized block coordinate selection strategy. This confines the optimization to a "selected set" of size $r$ (which is a parameter that depends on the given data set, and usually is much less than $n$), thereby reducing the complexity to $\mathbf{O}(r^3)$. In addition, unlike prior acceleration methods often based on constraint relaxation, our method can guarantee the strict feasibility through a novel "marginal compensation mechanism" to synchronize local mass redistribution with global constraints. Finally, we conduct a set of experiments on various datasets, and the results demonstrate that our method achieves a $5.8\times$ speedup on large-scale graph alignment benchmarks, while maintaining state-of-the-art accuracy.

## 1. Introduction

The **Gromov-Wasserstein (GW)** distance (Mémoli, 2011)

enables distribution comparison by matching the intrinsic geometry of structured data without relying on a shared embedding space. This embedding-agnostic property allows it to couple data with heterogeneous topological structures, making it widely adopted in tasks such as graph alignment (Gao et al., 2021; Tang et al., 2023), shape matching (Mémoli, 2009; Schmitzer & Schnörr, 2013), molecular structure comparison (Vayer et al., 2020; Wei et al., 2025), single-cell multi-omics integration (Demetci et al., 2022; Klein et al., 2024), and cross-modal generative modeling (Bunne et al., 2019).

Although the GW distance has gained widespread attention in structured data analysis and machine learning, its practical application remains hindered by high computational cost. This bottleneck arises from the need to evaluate the consistency between pairwise distance structures across two spaces (Mémoli, 2011). For example, if the number of samples in each domain is $n$, then each structure contains $\Theta(n^2)$ sample pairs, and a complete comparison of their relational patterns incurs a computational complexity as large as $\Theta(n^4)$. To address this challenge, several methods employ entropic regularization (Cuturi, 2013; Solomon et al., 2016; Le et al., 2022; Rioux et al., 2024; Zhang et al., 2024b), which relaxes the discrete coupling into a soft assignment. Crucially, this relaxation enables the use of the Sinkhorn algorithm (Cuturi, 2013), which solves the GW problem via efficient iterative matrix-vector multiplications, bypassing the computationally expensive combinatorial assignment solver (Kuhn, 1955; Munkres, 1957). Through this regularization, the complexity is reduced to $\mathbf{O}(n^3)$. However, the cubic complexity still limits its application in large-scale scenarios.

To further accelerate the computation for GW, most existing methods rely on strategies such as constraint relaxation or low-dimensional approximation (e.g., hierarchical coarsening, spectral truncation, and low-rank parameterization). BPG (Xu et al., 2019b) adopts an inexact proximal point framework, where the per-iteration computational cost consists of $\mathbf{O}(n^3)$ gradient computation and $\mathbf{O}(n^2)$ Sinkhorn iterations. To compress the inner loop, BPG-S (Xu et al., 2019b) simplifies the Sinkhorn solver to a single projection, while KL-BAPG (Li et al., 2023a) efficiently decouples the doubly stochastic constraints via an alternating single-step

[1]School of Computer Science and Technology, University of Science and Technology of China, Hefei, China [2]Department of Computer Science, City University of Hong Kong, Hong Kong, China [3]Shandong University, Jinan, China. Correspondence to: Hu Ding <huding@ustc.edu.cn>.

*Proceedings of the 43rd International Conference on Machine Learning*, Seoul, South Korea. PMLR 306, 2026. Copyright 2026 by the author(s).

projection strategy. However, these methods primarily optimize the relatively low-complexity projection step and do not resolve the $\mathbf{O}(n^3)$ bottleneck for gradient computation.

In contrast, the low-dimensional approximation methods bypass that $\mathbf{O}(n^3)$ bottleneck by reducing data dimensionality. Specifically, ScalaGW (Xu et al., 2019a) and HOT (Zeng et al., 2024) employ multi-scale coarsening and hierarchical decomposition strategies, respectively, to reduce the scale for the problem. Nevertheless, the performance of such hierarchical methods is often compromised by error accumulation, where the alignment quality in subsequent stages is heavily contingent upon the coarse matches established in earlier phases. Alternatively, relying on an assumption of spectral similarity, SpecGW (Chowdhury & Needham, 2021) utilizes spectral embeddings of the Laplacian operator to project graph structures onto a low-dimensional eigenbasis. Similarly, predicated on a low-rank assumption, LR (Scetbon et al., 2022) employs a dual low-rank parameterization for both the coupling matrix and the input distance matrices. Although these low-dimensional approximation strategies significantly reduce computational complexity, their effectiveness is strictly contingent upon strong structural priors. These limitations highlight the need for a scalable GW solver that reconciles computational efficiency with geometric fidelity.

## 1.1. Our Main Contributions

Motivated by the pros and cons of the aforementioned GW methods, we consider a fundamental question:

*Is it possible to design an effective GW algorithm, which is able to tackle the tricky $\mathbf{O}(n^3)$ per-iteration complexity, and meanwhile, guarantee the strict feasibility of the obtained solution without relaxation?*

To solve the above problems, we propose an efficient GW optimization algorithm via localized block coordinate descent, named *LoBCD-GW*. By integrating the localized block coordinate selection strategy with the feasibility-preserving marginal compensation mechanism, our method breaks the trade-off between accuracy and efficiency.

Motivated by the efficacy of *block coordinate descent (BCD)* in quadratic programming (Hsieh et al., 2008; Nutini et al., 2015; Xu et al., 2019b), we extend this paradigm to GW optimization. Technically, applying BCD to the GW problem presents a significant challenge, as the block coordinate selection strategy remains the key bottleneck governing algorithmic efficiency. For instance, consider a naive cyclic or random block selection strategy that partitions the coupling matrix into fixed-size sub-blocks. Even under this simplified scheme, which ignores the higher complexity of coupled updates for feasibility, updating all $\mathbf{O}(n^2)$ blocks with efficient incremental updates (each costing $\mathbf{O}(n^2)$) incurs a

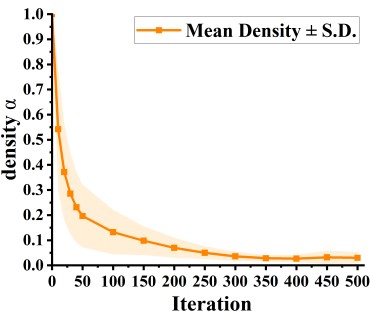

*Figure 1.* Evolution of the significant update density $\alpha$ produced by LoBCD-GW on the Reddit dataset, averaged over 500 graph pairs. Shaded regions denote the standard deviation. Here, $\alpha$ is defined as the proportion of coupling entries whose variation between consecutive iterations exceeds a threshold (set to $2.5 \times 10^{-6}$ for this visualization).

prohibitive $\mathbf{O}(n^4)$ cost per epoch, theoretically exceeding the $\mathbf{O}(n^3)$ cost of a standard full-batch GW iteration. To address this, we exploit the **data-dependent sparsity** inherent in GW optimization. As shown in Figure 1, *most coupling entries stabilize rapidly, thereby confining significant updates to a shrinking subset of sample pairs.* This indicates that global updates are computationally inefficient, as calculations involving these stabilized entries yield negligible optimization gains. A detailed analysis of coupling update sparsity is provided in Section 3.2. By confining the computationally intensive updates to a selected set which is identified by large constrained gradient magnitudes, we reduce the per-iteration complexity from $\mathbf{O}(n^3)$ to $\mathbf{O}(r^3)$, where $r \ll n$ is the cardinality of the selected set. This method achieves high efficiency while preserving the full geometric structure, avoiding the data loss common in methods that simplify the graph. Furthermore, unlike constraint relaxation methods, we propose a novel "*marginal compensation mechanism*" for localized block coordinate descent that guarantees strict feasibility. Finally, we prove the convergence of our algorithm and demonstrate superior efficiency and state-of-the-art accuracy on large-scale graph benchmarks.

## 2. Preliminaries

Due to the space limit, we leave the full discussion on related works in Section A. Below, we introduce several important notations and definitions.

**Some Notations.** Let $n$ and $m$ denote the number of samples in the source and target domains, respectively. We use $[n] \coloneqq \{1, 2, \ldots, n\}$ as the index set, and let $\mathbf{1}_n \in \mathbb{R}^n, \mathbf{1}_m \in \mathbb{R}^m$ be the all-ones vectors. For two discrete probability vectors $\mu \in \mathbb{R}_+^n$ and $\nu \in \mathbb{R}_+^m$ with $\mathbf{1}_n^\top \mu = \mathbf{1}_m^\top \nu = 1$, we define the set of **coupling matrices** as $\Pi(\mu, \nu) = \{\pi \in \mathbb{R}_+^{n \times m} \mid \pi \mathbf{1}_m = \mu, \pi^\top \mathbf{1}_n = \nu\}$, which represents the set of all joint distributions with marginals $\mu$ and $\nu$.

**Gromov–Wasserstein Distance.** In the discrete case, let $(X, D_s, \mu)$ and $(Y, D_t, \nu)$ be two discrete metric measure spaces, where $X = \{x_i\}_{i=1}^n$ and $Y = \{y_j\}_{j=1}^m$ denote the source and target sample sets, equipped with their intra-domain distance matrices $D_s \in \mathbb{R}^{n \times n}$ and $D_t \in \mathbb{R}^{m \times m}$, respectively. For simplicity, we assume uniform sample distributions $\mu = \frac{1}{n}\mathbf{1}_n$ and $\nu = \frac{1}{m}\mathbf{1}_m$. The discrete GW distance between $\mu$ and $\nu$ is defined as:

$$d_{GW}^2(\mu, \nu) = \min_{\pi \in \Pi(\mu,\nu)} \sum_{i,i',j,j'} |D_s(i,i') - D_t(j,j')|^2 \pi_{ij} \pi_{i'j'}$$

$$\text{s.t.} \quad \pi\mathbf{1}_m = \mu, \quad \pi^\top \mathbf{1}_n = \nu, \quad \pi \geq 0, \tag{1}$$

where $i, i' \in [n]$ and $j, j' \in [m]$ denote the indices of source and target samples, respectively. The constraints in Equation (1) ensure that $\pi$ is a valid joint distribution between the two spaces: the marginal constraints "$\pi\mathbf{1}_m = \mu$" and "$\pi^\top \mathbf{1}_n = \nu$" ensure that $\pi$ has marginals $\mu$ and $\nu$, respectively, while "$\pi \geq 0$" ensures that each entry $\pi_{ij}$ is non-negative, representing the matching score between source sample $x_i$ and target sample $y_j$. Once the optimized coupling $\pi^*$ is obtained, let $\mathcal{M}$ denote the set of all feasible point-to-point (injective) mappings between the source and target domains. Then, a discrete alignment $M^* \in \mathcal{M}$ can be derived by:

$$M^* = \arg\max_{M \in \mathcal{M}} \sum_{(x_i, y_j) \in M} \pi_{ij}^*. \tag{2}$$

Intuitively, the GW distance minimizes the total discrepancy between intra-domain pairwise distances. As shown in Figure 2, similar intra-domain distances $D_s(i, i')$ and $D_t(j, j')$ reduce the GW objective, thereby encouraging the optimal coupling to align $x_i \leftrightarrow y_j$ and $x_{i'} \leftrightarrow y_{j'}$ as part of a globally consistent mapping.

The quaternary summation in Equation (1) provides an intuitive illustration of the structural consistency inherent in the GW objective, but its high-order indexing obscures the underlying computational efficiency. To elucidate the algebraic structure, we specifically introduce the fourth-order loss tensor $\mathcal{C}$. By adopting the squared Euclidean loss $\ell(a, b) = (a - b)^2$, the elements of this tensor are defined as $\mathcal{C}_{iji'j'} := \ell(D_s(i, i'), D_t(j, j'))$. Furthermore, let $\otimes$ represent the tensor-matrix contraction operation defined by $(\mathcal{C} \otimes \pi)_{ij} := \sum_{i',j'} \mathcal{C}_{iji'j'} \pi_{i'j'}$, and $\langle \cdot, \cdot \rangle$ denote the Frobenius inner product. Consequently, the GW objective can be reformulated into the following equivalent matrix inner product form (Peyré et al., 2016):

$$\min_{\pi \in \Pi(\mu,\nu)} L(\pi) := \langle \mathcal{C}(D_s, D_t) \otimes \pi, \pi \rangle, \tag{3}$$

where $L(\pi)$ denotes the structural cost of the coupling $\pi$.

To perform a first-order update of $\pi$, we compute the gradi-

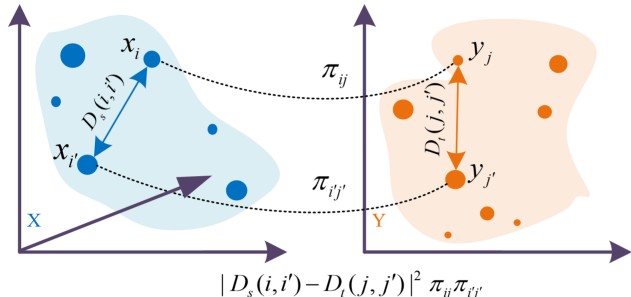

*Figure 2.* Geometric intuition of GW: Structurally consistent couplings.

ent of the objective function (3) as:

$$\nabla L(\pi) = u\mathbf{1}_m^\top + \mathbf{1}_n v^\top - 2D_s \pi D_t^\top, \tag{4}$$

where $u = (D_s \odot D_s)\mu \in \mathbb{R}^n$ and $v = (D_t \odot D_t)\nu \in \mathbb{R}^m$ are vectors, with $\odot$ denoting the Hadamard product. While the terms involving $u$ and $v$ can be precomputed or updated in $\mathbf{O}(n^2)$, the term "$D_s \pi D_t^\top$" remains the primary computational bottleneck with a complexity of $\mathbf{O}(n^3)$.

## 3. Our *LoBCD-GW* Framework

In this section, we introduce the *LoBCD-GW* framework to efficiently minimize the GW objective formulated in Equation (3). Specifically, in Section 3.1, we first introduce an important "data-dependent sparsity" perspective for analyzing the optimization of the GW objective. We adopt the BCD framework, which decomposes the coupling matrix into coordinate blocks to reduce the gradient complexity in Equation (4). Moreover, we propose a data-dependent localized block coordinate selection strategy, followed by a detailed theoretical analysis of sparsity in Section 3.2. In Section 3.3, we propose a marginal compensation mechanism to implement the localized block coordinate descent introduced in Section 3.1. This mechanism synchronizes the localized mass distribution with the global marginal constraints to ensure feasibility during the BCD process. Finally, we present the convergence analysis of the algorithm.

### 3.1. Sparsity and Localized Block Coordinate Selection

To ground our localized block coordinate selection, we first introduce the global optimization scheme for the GW objective in Equation (3). Specifically, we employ mirror descent induced by the Kullback–Leibler (KL) divergence (Beck & Teboulle, 2003; Bolte et al., 2014). At iteration $k$, utilizing the gradient derived in Equation (4), we solve the KL-proximal subproblem:

$$\pi^{(k+1)} = \arg\min_{\pi \in \Pi(\mu,\nu)} \left\{ \langle \nabla L(\pi^{(k)}), \pi \rangle + \frac{1}{\tau} \text{KL}(\pi || \pi^{(k)}) \right\}. \tag{5}$$

Given a **step size** $\tau$, the Karush–Kuhn–Tucker (KKT) conditions (Giorgi & Kjeldsen, 2013) of Equation (5) yield the closed-form multiplicative update rule (see Appendix B for the detailed derivation):

$$\pi^{(k+1)} = \mathrm{Proj}_{\Pi(\mu,\nu)}^{\mathrm{KL}}\left(\pi^{(k)} \odot \exp\left(-\tau \cdot \nabla L(\pi^{(k)})\right)\right). \quad (6)$$

Here, $\mathrm{Proj}_{\Pi(\mu,\nu)}^{\mathrm{KL}}$ is the projection onto the transport polytope with respect to the KL divergence, which can be efficiently implemented via the Sinkhorn algorithm (Cuturi, 2013).

In the optimization process defined by Equation (6), the global computation of the coupling matrix constitutes the primary computational bottleneck associated with the gradient evaluation in Equation (4). To address this, we propose a localized block coordinate selection strategy. Specifically, our strategy is based on an important observation of data-dependent sparsity in coupling matrix updates: *projected gradients confine significant updates to a shrinking subset of sample pairs, while the majority of entries rapidly stabilize.* This implies that performing global computations on these stabilized entries is largely redundant. A detailed analysis of this sparsity is provided in Section 3.2.

Let $\epsilon_\Delta > 0$ be a selection threshold that distinguishes stationary regions from those undergoing significant gradient-driven updates. We define the **selected set** $\mathcal{A}_k$ at iteration $k$ as:

$$\mathcal{A}_k = \left\{ (i,j) \mid \left|[\Delta \pi^{(k)}]_{ij}\right| > \epsilon_\Delta \right\}, \quad (7)$$

where the increment matrix $\Delta \pi^{(k)}$ is the change induced by a single projected mirror-descent step:

$$\Delta \pi^{(k)} = \mathrm{Proj}_{\Pi(\mu,\nu)}^{\mathrm{KL}}\left(\pi^{(k)} \odot \exp\left(-\tau \nabla L(\pi^{(k)})\right)\right) - \pi^{(k)}. \quad (8)$$

After a one-time full projected mirror-descent step for initializing the selected set, subsequent updates are performed within a localized block coordinate descent framework. Specifically, we restrict the optimization process to the coordinates within the selected set $\mathcal{A}_k$, while entries in the complement set $\mathcal{A}_k^c$ remain fixed:

$$\pi_{\mathcal{A}_k}^{(k+1)} \in \underset{\pi_{\mathcal{A}_k}}{\arg\min}\, L\left(\pi_{\mathcal{A}_k}, \pi_{\mathcal{A}_k^c}^{(k)}\right), \text{ s.t. } \pi_{\mathcal{A}_k^c}^{(k+1)} = \pi_{\mathcal{A}_k^c}^{(k)}, \quad (9)$$

where "$L(\cdot)$" is the objective function defined in Equation (3). Note that $\mathcal{A}_k$ is dynamically updated, where previously excluded entries can be re-selected if updates in the selected set perturb the gradient $\nabla L(\pi)$ through the interaction term $D_s \pi D_t^\top$, pushing their increments back above $\epsilon_\Delta$. Let

$$I_k = \{i : \exists j, (i,j) \in \mathcal{A}_k\},\ J_k = \{j : \exists i, (i,j) \in \mathcal{A}_k\}. \quad (10)$$

Consequently, the global optimization problem is reduced to an $r \times r$ local subspace spanned by $(I_k, J_k)$, where $r = \max\{|I_k|, |J_k|\}$.

## 3.2. Theoretical Analysis of Sparsity

Although the coupling matrix $\pi$ naively contains $\mathbf{O}(n^2)$ entries, we restrict updates to the selected set $\mathcal{A}_k$. In this section, we analyze the data-dependent sparsity of coupling-matrix updates and derive a bound on the size of $\mathcal{A}_k$, thereby providing a theoretical justification for the effectiveness of our selection strategy.

**Theorem 3.1** (Sparsity of Numerical Updates). *Let $\{\pi^{(k)}\}$ be a sequence converging to an optimal coupling $\pi^*$. Suppose $\pi^*$ satisfies the strict complementarity condition (Definition 3.2). For any selection threshold $\epsilon_\Delta > 0$, there exists an iteration index $K$ such that for all $k \geq K$,*

$$|\mathcal{A}_k| = \mathbf{O}(n). \quad (11)$$

Theorem 3.1 shows that the selected set $\mathcal{A}_k$ is reduced from the potential $\mathbf{O}(n^2)$ scale to $\mathbf{O}(n)$. Equation (10) specifies the row and column index sets $I_k$ and $J_k$ covered by $\mathcal{A}_k$. Barring the pathological worst case in which these $\mathbf{O}(n)$ selected entries are dispersed over $\Theta(n)$ distinct rows or columns, since $r = \max\{|I_k|, |J_k|\}$, we have $r \ll n$.

We provide the proof of Theorem 3.1 below. To establish this result, we first connect the KL projection in Equation (6) with the KKT conditions of the transport polytope, which yields an explicit form of the reduced gradients. We then introduce the standard non-degeneracy condition of strict complementarity (Definition 3.2), which serves as the basis for Lemma 3.4. Based on the foundations laid above, we finally provide the formal proof of Theorem 3.1.

**KL projection and reduced gradients.** Recall the mirror-descent step before projection,

$$\widetilde{\pi}^{(k+1)} = \pi^{(k)} \odot \exp\left(-\tau \nabla L(\pi^{(k)})\right). \quad (12)$$

The KL projection $\mathrm{Proj}_{\Pi(\mu,\nu)}^{\mathrm{KL}}(\cdot)$ can be implemented by Sinkhorn scaling, i.e., there exist positive vectors $a^{(k)} \in \mathbb{R}_+^n$ and $b^{(k)} \in \mathbb{R}_+^m$ such that

$$\pi^{(k+1)} = \mathrm{diag}(a^{(k)})\, \widetilde{\pi}^{(k+1)}\, \mathrm{diag}(b^{(k)}),$$
$$\text{s.t.} \quad \pi^{(k+1)} \mathbf{1}_m = \mu, \quad (\pi^{(k+1)})^\top \mathbf{1}_n = \nu. \quad (13)$$

Equivalently, by defining the (scaled) dual potentials

$$\alpha_i^{(k)} = -\frac{1}{\tau} \log a_i^{(k)}, \qquad \beta_j^{(k)} = -\frac{1}{\tau} \log b_j^{(k)}, \quad (14)$$

the projected update admits an entrywise form in terms of the *reduced gradient* $G_{ij}^{(k)}$ induced by the marginal constraints:

$$\pi_{ij}^{(k+1)} = \pi_{ij}^{(k)} \exp\left(-\tau\, G_{ij}^{(k)}\right), \quad (15)$$

where the reduced gradient is defined as:

$$G_{ij}^{(k)} := [\nabla L(\pi^{(k)})]_{ij} + \alpha_i^{(k)} + \beta_j^{(k)}. \quad (16)$$

In matrix form, the reduced gradient is given by:

$$G^{(k)} := \nabla L(\pi^{(k)}) + \alpha^{(k)} \mathbf{1}_m^\top + \mathbf{1}_n (\beta^{(k)})^\top \in \mathbb{R}^{n \times m}, \quad (17)$$

where $\alpha^{(k)}$ and $\beta^{(k)}$ are the corresponding dual potential vectors.

**Definition 3.2** (Strict Complementarity with Margin $\delta$). Let $\pi^* \in \Pi(\mu, \nu)$ be an optimal coupling and $S^* := \{(i, j) : \pi_{ij}^* > 0\}$ be its optimal support. Let $\alpha^* \in \mathbb{R}^n$ and $\beta^* \in \mathbb{R}^m$ be the dual potentials associated with the marginal constraints at $\pi^*$. We say $\pi^*$ satisfies *strict complementarity with margin* $\delta$ if the reduced gradient defined as

$$G_{ij}^* = [\nabla L(\pi^*)]_{ij} + \alpha_i^* + \beta_j^* \quad (18)$$

satisfies exactly one of the following conditions for all $(i, j) \in [n] \times [m]$:

1. On-support: if $(i, j) \in S^*$, then $G_{ij}^* = 0$;
2. Off-support: if $(i, j) \notin S^*$, then $G_{ij}^* \geq \delta > 0$.

*Remark* 3.3 (Geometric Intuition). Strict complementarity precludes degenerate optima where unmatched pairs yield zero reduced cost. In graph alignment terms, every matched pair is supported by a clear structural rationale, while every non-match is separated by a positive margin $\delta$, creating a geometric gap between the true correspondence region and the rest.

**Lemma 3.4** (Linear-Scale Optimal Support). *Under the data-dependent Candidate-Locality Assumption (i.e., for each source sample $i$, there exists a candidate set $\Omega(i)$ such that $\pi_{ij}^* > 0 \implies j \in \Omega(i)$ and $|\Omega(i)| \leq \Omega_{cand}$, where $\Omega_{cand}$ is a uniform bound on the number of target candidates per source sample), the optimal support size is bounded linearly by the number of source samples:*

$$|S^*| = \sum_{i=1}^n |\{j : \pi_{ij}^* > 0\}| \leq \sum_{i=1}^n |\Omega(i)| \leq n \cdot \Omega_{cand}.$$

*Proof.* (**of Theorem 3.1**) From Equation (15), the projected increment obeys

$$\pi_{ij}^{(k+1)} - \pi_{ij}^{(k)} = \pi_{ij}^{(k)} \left( \exp(-\tau G_{ij}^{(k)}) - 1 \right). \quad (19)$$

Consider an off-support index $(i, j) \notin S^*$. By Definition 3.2, we have $G_{ij}^* \geq \delta$. Since $\pi^{(k)} \to \pi^*$ and the associated Sinkhorn dual potentials converge along the iterates, we obtain $G_{ij}^{(k)} \to G_{ij}^*$, hence there exists $K$ such that for all $k \geq K$,

$$G_{ij}^{(k)} \geq \delta/2. \quad (20)$$

Plugging Equation (20) into Equation (15) yields geometric decay:

$$\pi_{ij}^{(k+1)} = \pi_{ij}^{(k)} \exp(-\tau G_{ij}^{(k)}) \leq \pi_{ij}^{(k)} \exp(-\tau \delta/2).$$

Moreover, since $\left| \exp(-\tau G_{ij}^{(k)}) - 1 \right| \leq 1$ for $G_{ij}^{(k)} \geq 0$, Equation (19) implies

$$\left| \pi_{ij}^{(k+1)} - \pi_{ij}^{(k)} \right| \leq \pi_{ij}^{(k)}, \quad (21)$$

and therefore the update magnitudes on off-support entries also decay geometrically. Hence, for any $\epsilon_\Delta > 0$, all off-support indices eventually satisfy $\left| \pi_{ij}^{(k+1)} - \pi_{ij}^{(k)} \right| \leq \epsilon_\Delta$, which implies $\mathcal{A}_k \subseteq S^*$ for all $k \geq K$. Consequently, invoking Lemma 3.4, the cardinality of the selected set is bounded linearly:

$$|\mathcal{A}_k| \leq |S^*| \leq n \cdot \Omega_{cand} = \mathbf{O}(n). \quad (22)$$

$\square$

### 3.3. Marginal Compensation for Localized Block Coordinate Descent

In this section, we introduce a feasibility-preserving *marginal compensation mechanism* to maintain marginal constraints while updating the coupling matrix $\pi$ within the selected sub-block spanned by $(I_k, J_k)$ defined in (10), as detailed in Algorithm 1. The core idea is to redistribute the global marginal requirements by subtracting the mass already assigned to stationary entries outside the selected sub-block.

Specifically, for a selected sub-block $\pi_{\text{sub}} = \pi[I_k, J_k]$, we derive the local marginals $\hat{\mu}$ and $\hat{\nu}$ as follows:

$$\hat{\mu}_i = \mu_i - \sum_{j \notin J_k} \pi_{ij}^{(k)}, \forall i \in I_k; \quad \hat{\nu}_j = \nu_j - \sum_{i \notin I_k} \pi_{ij}^{(k)}, \forall j \in J_k. \quad (23)$$

**Lemma 3.5** (Feasibility Preservation). *Let $\pi^{(k+1)}$ be the global coupling matrix obtained by embedding the updated $\pi_{\text{sub}}$ back into the full matrix $\pi^{(k)}$. If $\pi_{\text{sub}}$ satisfies the local marginal constraints with respect to $\hat{\mu}$ and $\hat{\nu}$, then $\pi^{(k+1)}$ strictly satisfies the global marginal constraints $\Pi(\mu, \nu)$.*

*Proof.* For any row index $i \in I_k$, the total mass in $\pi^{(k+1)}$ can be decomposed as:

$$\sum_{j=1}^m \pi_{ij}^{(k+1)} = \sum_{j \in J_k} (\pi_{\text{sub}})_{ij} + \sum_{j \notin J_k} \pi_{ij}^{(k)}. \quad (24)$$

Substituting the definition of $\hat{\mu}_i$ from Equation (23), we obtain:

$$\sum_{j=1}^m \pi_{ij}^{(k+1)} = \hat{\mu}_i + (\mu_i - \hat{\mu}_i) = \mu_i. \quad (25)$$

Analogously, $\sum_{i=1}^n \pi_{ij}^{(k+1)} = \nu_j$ for $j \in J_k$. Since $\pi_{\text{sub}} \geq 0$ via Sinkhorn projection and stationary entries are non-negative, we conclude $\pi^{(k+1)} \in \Pi(\mu, \nu)$. $\square$

**Algorithm 1** Local Block Update

1: **Input:** Global coupling $\pi^{(k)}$, selected indices $(I_k, J_k)$
2: **Output:** Optimal local coupling matrix $\pi^*_{\text{sub}}$
3: **Step one: Marginal Compensation**
4: **for** $i \in I_k$ **do**
5:    $\hat{\mu}_i \leftarrow \mu_i - \sum_{j \notin J_k} \pi^{(k)}_{ij}$
6: **end for**
7: **for** $j \in J_k$ **do**
8:    $\hat{\nu}_j \leftarrow \nu_j - \sum_{i \notin I_k} \pi^{(k)}_{ij}$
9: **end for**
10: **Step two: Localized computation**
11: Initialize $\pi^{(0)}_{\text{sub}} \leftarrow \pi^{(k)}[I_k, J_k]$
12: $s \leftarrow 0$
13: **repeat**
14:    $\mathbf{M}_{\text{sub}} \leftarrow \mathbf{D}_s[I_k, I_k]\pi^{(s)}_{\text{sub}}\mathbf{D}_t[J_k, J_k]^\top$
15:    $\nabla L(\pi^{(s)})_{\text{sub}} = u_{\text{sub}}\mathbf{1}^\top + \mathbf{1}v_{\text{sub}}^\top - 2M_{\text{sub}}$
16:    $\pi^{(s+1)}_{\text{sub}} \leftarrow \text{Proj}^{\text{KL}}_{\Pi(\hat{\mu}, \hat{\nu})}\Big(\pi^{(s)}_{\text{sub}} \odot \exp\big(-\tau \nabla L(\pi^{(s)})_{\text{sub}}\big)\Big)$
17:    $s \leftarrow s + 1$
18: **until** local convergence
19: **return** $\pi^*_{\text{sub}}$

**Algorithm 2** Localized Block Coordinate GW

1: **Input:** Source and target distance matrices $D_s, D_t$, marginals $\mu, \nu$, selected threshold $\epsilon_\Delta$, and step size $\tau$.
2: **Output:** Optimal coupling matrix $\pi^*$
3: **Initialize:** $\pi^{(0)}, k \leftarrow 0$
4: **repeat**
5:    Step 1: Localized block coordinate selection.
6:    Form selected set $\mathcal{A}_k = \{(i,j) : |\Delta\pi^{(k)}_{ij}| > \epsilon_\Delta\}$
7:    Extract indices $I_k, J_k$ from selected set $\mathcal{A}_k$
8:    Step 2: Local Block Update.
9:    $\pi^*_{\text{sub}} \leftarrow$ Algorithm 1
10:    Step 3: Update global solution.
11:    $M_{\mathcal{A}_k} \leftarrow M_{\text{sub}}$
12:    $\pi^{(k+1)}_{\mathcal{A}_k} \leftarrow \pi^*_{\text{sub}}$
13:    $\pi^{(k+1)}_{\mathcal{A}^c_k} = \pi^{(k)}_{\mathcal{A}^c_k}$
14:    $k \leftarrow k + 1$
15: **until** global convergence of $\pi^{(k)}$ is reached (e.g., $\|\pi^{(k)} - \pi^{(k-1)}\|_{\text{F}} < \epsilon_{\text{global}}$)
16: **return** $\pi^{(k)}$

**Localized computation.** With feasibility guaranteed by Lemma 3.5, the KL-proximal update (6) can be restricted to the selected sub-block:

$$\pi^{(s+1)}_{\text{sub}} \leftarrow \text{Proj}^{\text{KL}}_{\Pi(\hat{\mu}, \hat{\nu})}\Big(\pi^{(s)}_{\text{sub}} \odot \exp\big(-\tau \nabla L(\pi^{(s)})_{\text{sub}}\big)\Big) \quad (26)$$

where the localized gradient $\nabla L(\pi^{(s)})_{\text{sub}}$ is defined as:

$$\nabla L(\pi^{(s)})_{\text{sub}} = u_{\text{sub}}\mathbf{1}^\top + \mathbf{1}v_{\text{sub}}^\top - 2M_{\text{sub}}, \quad (27)$$

and the interaction term $M_{\text{sub}}$ is efficiently computed via sub-matrix multiplication:

$$M_{\text{sub}} = D_s[I_k, I_k]\pi^{(s)}_{\text{sub}}D_t[J_k, J_k]^\top. \quad (28)$$

By integrating the localized selection strategy with the feasibility-preserving mechanism, the LoBCD-GW framework is implemented as detailed in Algorithm 2.

**Theorem 3.6** (Time complexity). *After the one-time initialization of the selected set, the LoBCD-GW algorithm has a per-iteration time complexity of* $\mathrm{O}(r^3)$. *This cost consists of* $\mathrm{O}(r^3)$ *for the interaction term computation* (28) *and* $\mathrm{O}(r^2)$ *for the localized Sinkhorn projection* (26), *where* $r \ll n$.

After establishing the LoBCD-GW framework, a fundamental question arises: whether LoBCD-GW converges. We answer affirmatively. Under standard assumptions, we demonstrate the sufficient decrease of $L(\pi)$ (Lemma 3.7), which ensures that the sequence of couplings eventually converges to a stationary point of the GW problem (Theorem 3.8).

**Lemma 3.7** (Sufficient Decrease). *Let $L_g$ denote the Lipschitz constant of the gradient $\nabla L(\pi)$ in* (4), *and let $c_{\text{KL}}$ denote the strong convexity modulus of the proximal term $\text{KL}(\cdot \| \pi^{(k)})$ defined in* (5). *If the step size satisfies $\tau < 2c_{\text{KL}}/L_g$, the iterates generated by LoBCD-GW satisfy:*

$$L(\pi^{(k+1)}) - L(\pi^{(k)}) \leq -\underbrace{\left(\frac{c_{\text{KL}}}{\tau} - \frac{L_g}{2}\right)}_{\rho > 0}\|\pi^{(k+1)} - \pi^{(k)}\|^2_F.$$
$$(29)$$

*Consequently, the objective function value $L(\pi)$ is monotonically non-increasing.*

To establish convergence, we utilize the *Global Proximal Gradient Mapping*, defined as:

$$G_\tau(\pi^{(k)}) = \frac{1}{\tau}\left(\pi^{(k)} - \pi^{(k+1)}\right). \quad (30)$$

**Theorem 3.8** (Global Convergence). *Given the sufficient decrease property established in Lemma 3.7, the global proximal gradient mapping of the iterates generated by LoBCD-GW satisfies:*

$$\lim_{k \to \infty} \|G_\tau(\pi^{(k)})\|_F = 0. \quad (31)$$

*This implies that every limit point of the sequence $\{\pi^{(k)}\}$ is a stationary point of the original GW problem.*

The proofs of Lemma 3.7 and Theorem 3.8 are deferred to Appendix D.

# 4. Experiments

In this section, we present a comprehensive evaluation to demonstrate the effectiveness of the proposed LoBCD-GW. Experiments are conducted on graph datasets of varying scales, focusing on the canonical task of ***graph alignment*** (Xu et al., 2019b; Tang et al., 2023; Li et al., 2023a). All experiments are implemented in Python 3.8 and executed on a high-performance server equipped with an Intel Xeon Platinum 8358P CPU and an NVIDIA A100-SXM4 GPU. All methods employ the same stopping criterion (following prior work on GW (Li et al., 2023a)): the optimization terminates if $\|\pi^{(k+1)} - \pi^{(k)}\|_F^2 / \|\pi^{(k)}\|_F^2 \leq 10^{-6}$ or the iteration index $k$ reaches 2,000.

## 4.1. Graph Alignment

Graph alignment seeks node-level correspondences between structurally related graphs (Zhang et al., 2021; Tang et al., 2023). We benchmark LoBCD-GW against several competitive baselines, categorized into two groups: (1) constraint relaxation methods, including eBPG (Solomon et al., 2016), BPG (Xu et al., 2019b) and its single-step variant BPG-S (Xu et al., 2019b), FW (Titouan et al., 2019a), KL-BAPG (Li et al., 2023a), and JOENA (Yu et al., 2025); (2) low-dimensional approximation methods, including ScalaGW (Xu et al., 2019a), SpecGW (Chowdhury & Needham, 2021), LR (Scetbon et al., 2022), and HOT (Zeng et al., 2024).

**Datasets.** We evaluate the effectiveness of LoBCD-GW on five benchmark graph datasets: one synthetic dataset (avg. 1500 nodes, 56579 edges) following the construction method of (Li et al., 2023a) and four real-world benchmark datasets with diverse topologies and scales, spanning two application domains (see Table 2): (1) social networks (*Reddit* (avg. 375.9 nodes, 499.3 edges) (Li et al., 2023a), *Douban-230* (avg. 234 nodes, 345.33 edges) (Zeng et al., 2024)) and (2) co-author networks (*ACM-1000* (avg. 1000 nodes, 4163 edges) (Zeng et al., 2024), *DBLP-1000* (avg. 1000 nodes, 3747 edges) (Zeng et al., 2024)). We report both **accuracy** and **efficiency**: the graph alignment accuracy is defined as $\text{Acc} = \frac{|\Gamma_{\text{pred}} \cap \Gamma_{\text{gt}}|}{|\Gamma_{\text{gt}}|} \times 100\%$, where $\Gamma_{\text{gt}}$ is the set of ground-truth node correspondences and $\Gamma_{\text{pred}}$ is the set produced by the algorithm; the efficiency is the average wall-clock time per graph pair (seconds).

**Parameter Setup.** We use symmetrically normalized, unweighted adjacency matrices as input distance matrices. The source and target marginals $\mu$ and $\nu$ are set to uniform. In the experiments, the step size $\tau$ is selected from $\{2, 5, 10, 20, 100\}$ to achieve optimal alignment performance. The selection threshold is set as $\epsilon_\Delta \in \{2.5 \times 10^{-6}, 2.5 \times 10^{-7}\}$. Baselines use the hyperparameters reported in their original articles (see Appendix E.8). To

ensure a fair comparison, all evaluated methods use the same initialization protocol. For each method, we report the better result obtained from two initialization strategies: (1) standard independent coupling, $\pi^{(0)} \leftarrow \mu\nu^\top$; and (2) degree-based heuristic initialization,

$$\pi^{(0)} \leftarrow \text{Proj}_{\Pi(\mu,\nu)}^{\text{KL}} \left( \exp\left( -\frac{\Delta_d}{\bar{\Delta}_d \cdot \text{temp} + \text{eps}} \right) \right),$$

where $\Delta_d = |D_s \mathbf{1} - (D_t \mathbf{1})^\top|$ and $\bar{\Delta}_d = \frac{1}{mn}\|D_s\mathbf{1} - (D_t\mathbf{1})^\top\|_1$. We use $\text{temp} = 0.1$ and $\text{eps} = 10^{-10}$ in all experiments.

**Ablation Studies.** Two variants are established: (1) w/o Local $M$ (using global $M$ instead of local updates), and (2) w/o Local Sink: (using a global Sinkhorn projection instead of the local Sinkhorn iterations $\text{Proj}_{\Pi(\hat{\mu},\hat{\nu})}^{\text{KL}}$ defined in Equation (26).

**Results and discussion.** Table 1 reports the alignment accuracy (%) and wall-clock time (s) across five datasets. We can see that LoBCD-GW achieves the highest accuracy on all datasets, significantly outperforming existing methods. For large-scale graphs (*Synthetic*, *ACM-1000*, and *DBLP-1000*), LoBCD-GW achieves substantially shorter runtimes than most baselines under the same hardware setting, especially for methods executed on GPU. For small graphs (*Douban-230* and *Reddit*), the overall computational time is already sub-second for several methods, so the runtime advantage of LoBCD-GW is less pronounced. Furthermore, the ablation studies (shown in Appendix E.1) demonstrate that the local Sinkhorn iterations and local $M$ updates significantly accelerate GW computation while maintaining the accuracy of graph alignment tasks.

## 4.2. Parameter Sensitivity Analysis

We conduct a comprehensive sensitivity analysis of LoBCD-GW to evaluate its robustness and stability with respect to key hyperparameters: the selection threshold $\epsilon_\Delta$ and the step size $\tau$. The experiments are performed on the same five datasets as in Section 4.1.

As shown in Figure 3(a), LoBCD-GW is insensitive to the selection threshold $\epsilon_\Delta$ on the Douban-230, Reddit, and Synthetic datasets. For $\epsilon_\Delta \in \{10^{-8}, 10^{-7}, 2.5 \times 10^{-7}, 5 \times 10^{-7}, 2.5 \times 10^{-6}\}$, accuracy varies by no more than 3%. By contrast, ACM-1000 and DBLP-1000 remain stable up to $\epsilon_\Delta = 2.5 \times 10^{-7}$ but exhibit a clear drop once $\epsilon_\Delta > 2.5 \times 10^{-7}$. These trends indicate that overly aggressive thresholds over-prune informative couplings, whereas small-to-moderate values (e.g., $\epsilon_\Delta \leq 2.5 \times 10^{-7}$) preserve high accuracy across datasets.

As shown in Figure 3(b), LoBCD-GW remains generally robust to changes in $\tau$. On the Synthetic dataset, the accuracy is flat at 100% across $\tau \in \{2, 5, 10, 20, 100\}$. Douban-230

*Table 1.* Comparison of alignment accuracy (%) and wall-clock time (seconds) across small-scale (Douban-230, Reddit) and large-scale (Synthetic, ACM-1000, DBLP-1000) datasets. Bold indicates the best performance. For fairness, each baseline is run in its natural implementation setting; the upper and lower blocks report results obtained on CPU and GPU, respectively. LoBCD-GW is evaluated on both CPU and GPU for fair comparison.

| Method | Douban-230 | | Reddit | | Synthetic | | ACM-1000 | | DBLP-1000 | |
|---|---|---|---|---|---|---|---|---|---|---|
| | Acc | Time | Acc | Time | Acc | Time | Acc | Time | Acc | Time |
| ScalaGW | 2.53 | 0.30 | 0.54 | 0.41 | 17.93 | 24.53 | 1.17 | 4.82 | 1.18 | 3.55 |
| FW | 34.00 | 0.11 | 21.51 | 6.31 | 24.50 | 27.29 | 88.90 | 80.87 | 74.53 | 122.36 |
| SpecGW | 66.08 | **0.03** | 50.71 | **0.19** | 13.27 | 4.23 | 93.73 | **0.50** | 86.40 | **0.53** |
| LoBCD-GW (CPU) | **99.71** | 0.52 | **97.18** | 0.57 | **100.0** | 1.14 | **99.03** | 0.63 | **97.30** | 0.65 |
| eBPG | 6.27 | 1.65 | 3.76 | 1.27 | 34.33 | 13.70 | 0.43 | 5.49 | 0.20 | 1.91 |
| BPG-S | 24.43 | 0.25 | 39.04 | 0.98 | 61.48 | 30.55 | 1.37 | 4.59 | 0.30 | 4.09 |
| BPG | 24.44 | **0.24** | 39.04 | 1.07 | 57.56 | 31.44 | 1.37 | 4.58 | 0.30 | 4.17 |
| LR | 66.50 | 0.33 | 1.76 | 14.75 | 7.24 | 5.16 | 41.00 | 7.10 | 16.13 | 3.73 |
| KL-BAPG | 68.19 | 0.66 | 50.93 | 0.23 | 99.79 | 4.18 | 94.00 | 1.46 | 86.40 | 1.42 |
| JOENA | 74.29 | 12.18 | 87.61 | 11.03 | 82.26 | 64.30 | 90.12 | 38.25 | 90.99 | 37.65 |
| HOT | 61.90 | 35.00 | 92.68 | 2.23 | **100.0** | 70.83 | 91.70 | 89.96 | 74.50 | 105.72 |
| LoBCD-GW (GPU) | **99.71** | 0.45 | **97.18** | 0.22 | **100.0** | 0.27 | **99.03** | 0.49 | **97.30** | 0.52 |

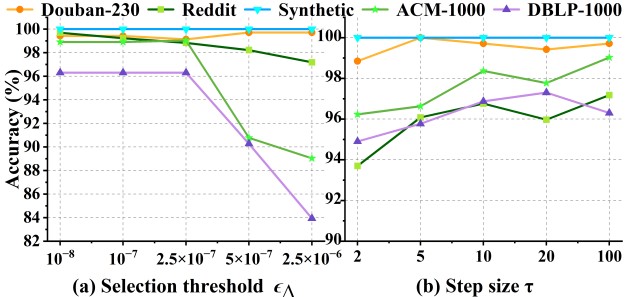

*Figure 3.* Sensitivity of LoBCD-GW to hyperparameters.

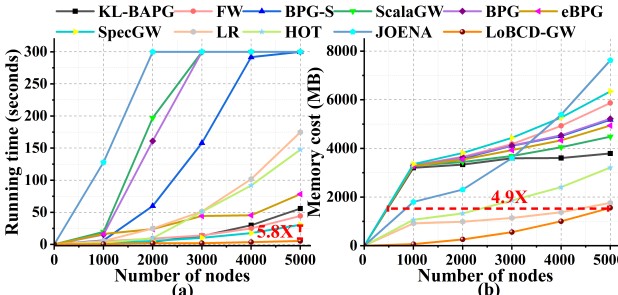

*Figure 4.* Scalability analysis of LoBCD-GW.

remains within the range of 98%–100% with only a mild decline when $\tau < 5$. ACM-1000 and Reddit are stable for $\tau > 10$ but degrade gradually when $\tau < 10$. DBLP-1000 peaks around $\tau = 20$ before decreasing slightly. These patterns suggest that a larger step size (e.g., $\tau \in [10, 100]$) yields consistently strong performance across datasets.

These findings support the hyperparameter robustness of our method and suggest that LoBCD-GW does not require too much hyperparameter tuning, which enhances its practicality.

### 4.3. Scalability Analysis

To evaluate scalability, we follow the synthesis protocol in (Xu et al., 2019b) and generate datasets with 20% noise across node scales $n \in \{1000, 2000, 3000, 4000, 5000\}$, subject to a 300-second timeout. Regarding computational efficiency (Figure 4(a)), LoBCD-GW maintains the lowest runtime across all scales. Table 3 lists the per-scale

runtimes of LoBCD-GW, and Figure 5 further visualizes the runtime trends on a log scale. As the number of nodes increases, LoBCD-GW achieves up to a $5.8\times$ speedup compared to the baselines. This efficiency gain stems from the localized block coordinate descent of LoBCD-GW. In contrast, baseline methods, hampered by the dense $\mathbf{O}(n^3)$ bottleneck, reach the timeout limit as $n$ scales to 2000. Regarding memory efficiency (Figure 4(b)), LoBCD-GW significantly alleviates storage bottlenecks. As the number of nodes increases to 5000, it reduces peak memory consumption by $4.9\times$ compared to baseline methods. These results validate the superior scalability of LoBCD-GW on large-scale graphs.

### 4.4. Additional Experiments

To further demonstrate the robustness and broad applicability of our method, we present a series of supplemental experiments. Specifically, for graph alignment, we con-

duct a robustness analysis involving five independent trials with distinct random seeds under noisy conditions (see Appendix E.2) and provide additional baseline comparisons in Appendix E.3 for completeness. We also include feasibility and gradient analyses in Appendix E.4, together with a GW objective value analysis in Appendix E.5, to further examine the optimization behavior of LoBCD-GW. Furthermore, we extend LoBCD-GW to the frontier tasks of graph partitioning (see Appendix E.6) and single-cell multi-omics data integration (see Appendix E.7), showing its versatility across diverse applications.

## 5. Conclusions

In this work, we present an efficient algorithm to compute the GW distance. By leveraging data-dependent sparsity, our method breaks the cubic per-iteration computational bottleneck, enabling scalability to large-scale real-world tasks. Furthermore, LoBCD-GW can also be applied to scenarios that demand high structural integrity, as it achieves high-fidelity preservation of distance matrices by directly optimizing the standard GW objective and ensuring strict feasibility. For future work, we aim to further investigate the theoretical properties and broaden the applications of the GW framework. Overall, the proposed method holds significant potential for tackling large-scale distribution alignment problems across diverse machine learning applications.

## Acknowledgements

We thank the anonymous reviewers for their valuable comments and suggestions. This work was partially supported by the National Key Research and Development Program of China (No. 2021YFA1000900), the National Natural Science Foundation of China (Nos. 62272432 and 62432016), and the Natural Science Foundation of Anhui Province (No. 2208085MF163). The experiments were supported by the robotic AI-Scientist platform of the Chinese Academy of Sciences.

## Impact Statement

This paper presents work whose goal is to advance the field of Machine Learning. There are many potential societal consequences of our work, none which we feel must be specifically highlighted here.

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

## A. More Related Works

**Gromov-Wasserstein Distance.** The GW distance is a key metric in machine learning. It extends optimal transport (OT) (Monge, 1781; Kantorovich, 1942; Villani, 2008) from ground-metric-based matching to structure-preserving alignment. However, the GW distance computation entails solving an NP-hard quadratic assignment problem (Titouan et al., 2019b). Cuturi (2013) introduced the Sinkhorn distance, which incorporates entropic regularization to reformulate the OT problem as a matrix-scaling problem, enabling efficient computation and circumventing the prohibitive costs of exact solvers. Following the work of Cuturi, several methods have improved Sinkhorn iterations (Benamou et al., 2015; Altschuler et al., 2017; Lin et al., 2019; Altschuler et al., 2019; Li et al., 2023a), while another line of work reformulates the per-iteration GW subproblem as an entropically regularized OT problem (Solomon et al., 2016; Xu et al., 2019b). While these methods stabilize and accelerate GW optimization, their cubic complexity remains prohibitive for large-scale datasets. To this end, another category of methods reduces overhead by employing structural approximations of the cost and coupling matrices (Xu et al., 2019a; Zeng et al., 2024; Chowdhury & Needham, 2021; Scetbon et al., 2022). However, their effectiveness is often constrained by the dependence on specific structural assumptions.

**Block Coordinate Descent.** BCD is an optimization paradigm that partitions high-dimensional variables into blocks and iteratively updates the solution by solving a subproblem for a selected block while holding the remaining blocks fixed (Bertsekas, 1997). Hildreth (1957) pioneers the application of this method to convex quadratic programming. While early studies focus on convex settings, the convergence theory for nonconvex optimization is firmly established by the seminal works of Grippo and Sciandrone (2000) and Tseng (2001). Given that the GW problem is formally a specific nonconvex instance of quadratic programming (closely related to the quadratic assignment problem), these works serve as historical precursors to our method. To address scalability in high-dimensional regimes, Nesterov (2012) establishes global complexity bounds for randomized coordinate descent. However, the block selection strategy remains the critical bottleneck governing algorithmic efficiency. Nutini et al. (2015) demonstrate that greedy, data-dependent selection rules (e.g., the Gauss-Southwell rule) significantly accelerate convergence compared to cyclic or randomized schemes. Motivated by this insight, we propose the localized block coordinate descent (LoBCD) algorithm, which incorporates a data-dependent localized block coordinate selection strategy (detailed in Section 3.1).

## B. Derivation of the Multiplicative Update Rule

In this section, we provide the detailed derivation of the closed-form solution in Equation (6) from the KL-proximal subproblem defined in Equation (5).

Recall the optimization problem at iteration $k$:

$$\pi^{(k+1)} = \underset{\pi \in \Pi(\mu, \nu)}{\arg\min} \left\{ \langle \nabla L(\pi^{(k)}), \pi \rangle + \frac{1}{\tau} \text{KL}(\pi || \pi^{(k)}) \right\}, \tag{32}$$

where $\text{KL}(\pi || \pi^{(k)}) = \sum_{i,j} \pi_{ij} \log(\frac{\pi_{ij}}{\pi_{ij}^{(k)}}) - \sum_{i,j} \pi_{ij} + \sum_{i,j} \pi_{ij}^{(k)}$ is the generalized KL divergence. Since both $\pi$ and $\pi^{(k)}$ reside in the transport polytope $\Pi(\mu, \nu)$, the linear mass terms $\sum_{i,j} \pi_{ij}$ and $\sum_{i,j} \pi_{ij}^{(k)}$ cancel each other out. These terms are nevertheless retained to simplify the algebraic derivation of the gradient.

To solve (32), we introduce Lagrange multipliers $\alpha^{(k)} \in \mathbb{R}^n$ and $\beta^{(k)} \in \mathbb{R}^m$ for the row constraint $\pi \mathbf{1}_m = \mu$ and column constraint $\pi^\top \mathbf{1}_n = \nu$, respectively. The Lagrangian $\mathcal{J}(\pi, \alpha^{(k)}, \beta^{(k)})$ is given by:

$$\mathcal{J} = \sum_{i,j} [\nabla L(\pi^{(k)})]_{ij} \pi_{ij} + \frac{1}{\tau} \sum_{i,j} \left( \pi_{ij} \log \frac{\pi_{ij}}{\pi_{ij}^{(k)}} - \pi_{ij} + \pi_{ij}^{(k)} \right) + \sum_i \alpha_i^{(k)} (\sum_j \pi_{ij} - \mu_i) + \sum_j \beta_j^{(k)} (\sum_i \pi_{ij} - \nu_j). \tag{33}$$

Setting the partial derivative with respect to $\pi_{ij}$ to zero yields the first-order optimality condition:

$$\frac{\partial \mathcal{J}}{\partial \pi_{ij}} = [\nabla L(\pi^{(k)})]_{ij} + \frac{1}{\tau} \log \left( \frac{\pi_{ij}}{\pi_{ij}^{(k)}} \right) + \alpha_i^{(k)} + \beta_j^{(k)} = 0. \tag{34}$$

Rearranging the terms, we obtain the entrywise update rule:

$$\pi_{ij} = \pi_{ij}^{(k)} \exp \left( -\tau ([\nabla L(\pi^{(k)})]_{ij} + \alpha_i^{(k)} + \beta_j^{(k)}) \right). \tag{35}$$

By defining the *reduced gradient* as $G_{ij}^{(k)} := [\nabla L(\pi^{(k)})]_{ij} + \alpha_i^{(k)} + \beta_j^{(k)}$, the update is expressed as $\pi_{ij} = \pi_{ij}^{(k)} \exp(-\tau G_{ij}^{(k)})$. Letting $a_i^{(k)} = \exp(-\tau \alpha_i^{(k)})$ and $b_j^{(k)} = \exp(-\tau \beta_j^{(k)})$, we recover the dual potential relationships $\alpha_i^{(k)} = -\frac{1}{\tau} \log a_i^{(k)}$ and $\beta_j^{(k)} = -\frac{1}{\tau} \log b_j^{(k)}$. The update can thus be written in matrix form:

$$\pi^{(k+1)} = \text{diag}(a^{(k)}) \cdot \left( \pi^{(k)} \odot \exp(-\tau \nabla L(\pi^{(k)})) \right) \cdot \text{diag}(b^{(k)}). \tag{36}$$

Equation (36) confirms that the optimal solution $\pi^{(k+1)}$ is obtained via alternating scaling on the rows and columns of the mirror-descent step. This operation is precisely the Sinkhorn algorithm implementation of the KL projection onto the transport polytope. Therefore, we arrive at the final update rule:

$$\pi^{(k+1)} = \text{Proj}_{\Pi(\mu,\nu)}^{\text{KL}} \left( \pi^{(k)} \odot \exp \left( -\tau \nabla L(\pi^{(k)}) \right) \right). \tag{37}$$

## C. Structural Conditions for Linear-Scale Optimal Support

In this appendix, we provide the theoretical justification for the linear scaling bound $|S^*| = \text{O}(n)$ asserted in Lemma 3.4. We formally define the structural assumptions regarding *Candidate Locality* and prove the sparsity of the optimal support under the strict complementarity condition.

### C.1. Candidate Set and Counting Bound

For each source sample $i \in [n]$, let $\Omega(i) \subseteq [m]$ denote its set of admissible target candidates, and define the global candidate set as $\mathcal{P} = \{(i,j) : j \in \Omega(i)\}$. We assume a uniform row-wise bound on the candidate size:

$$|\Omega(i)| \leq \Omega_{\text{cand}}, \quad \forall i \in [n], \tag{38}$$

where $\Omega_{\text{cand}}$ is a constant independent of $n$ and $m$.

**Lemma C.1** (Counting Bound). *If the optimal support satisfies $S^* \subseteq \mathcal{P}$, then the total size of the support scales linearly with the sample size:*

$$|S^*| \leq \sum_{i=1}^{n} |\Omega(i)| \leq n \cdot \Omega_{cand} = \text{O}(n). \tag{39}$$

*Proof.* The proof follows directly by summing the candidate limits for each row. Since each source node $i$ contributes at most $|\Omega(i)|$ entries to the support, the global sparsity is bounded by $\Omega_{\text{cand}} n$. $\square$

The critical challenge is to justify the assumption $S^* \subseteq \mathcal{P}$. Below, we validate this under two common structural priors: $\kappa$-Nearest Neighbors ($\kappa$-NN) and Banded-Gap structure.

### C.2. Case 1: $\kappa$-Nearest Neighbors ($\kappa$-NN) Structure

In geometric graph alignment, true correspondences typically exist between nodes with similar local substructures. Let $\kappa$ denote the neighbor count parameter. For each source node $i$, let $\Omega_\kappa(i)$ be the set of its $\kappa$ nearest neighbors in the target space under a specific metric. The candidate set is defined as $\mathcal{P}_\kappa = \{(i,j) : j \in \Omega_\kappa(i)\}$.

To rigorously guarantee $S^* \subseteq \mathcal{P}_\kappa$, we invoke the optimality conditions of the GW problem. Recall the reduced gradient at optimality:

$$G_{ij}^* = [\nabla L(\pi^*)]_{ij} + \alpha_i^* + \beta_j^* \geq 0. \tag{40}$$

The strict complementarity condition states that $\pi_{ij}^* > 0 \implies G_{ij}^* = 0$. Conversely, if $G_{ij}^* > 0$, then $\pi_{ij}^* = 0$.

**Assumption C.2** ($\kappa$-NN Separation). We assume that with a sufficient neighbor count $\kappa$, the reduced gradient satisfies a separation margin $\delta > 0$ for all non-neighbor pairs:

$$G_{ij}^* \geq \delta > 0, \quad \forall j \notin \Omega_\kappa(i). \tag{41}$$

**Proposition C.3** ($\kappa$-NN Support Containment). *Under the $\kappa$-NN Separation assumption, the optimal support is contained within the candidate set ($S^* \subseteq \mathcal{P}_\kappa$), and consequently $|S^*| \leq \kappa n = \text{O}(n)$.*

*Proof.* Consider any pair $(i, j)$ with $j \notin \Omega_\kappa(i)$. By Assumption C.2, $G_{ij}^* \geq \delta > 0$. Strict complementarity then dictates that $\pi_{ij}^* = 0$. Therefore, all non-zero entries of the optimal coupling must satisfy $j \in \Omega_\kappa(i)$. This yields a constant row-wise bound of $|\Omega_\kappa(i)| = \kappa$. Thus, by Lemma C.1, we have $|S^*| = O(n)$. □

### C.3. Case 2: Banded-Gap Structure

For graph pairs whose nodes admit a natural 1-D ordering, we assume the nodes are indexed sequentially. In this setting, the optimal coupling matrix typically concentrates near the main diagonal. Let $W$ be a bandwidth parameter. The candidate set is defined as:

$$\mathcal{P}_W = \{(i, j) : |j - i| \leq W\}. \tag{42}$$

The size of this candidate set is bounded row-wise by $|\Omega_W(i)| \leq 2W + 1$, where $\Omega_W(i) = \{j : (i, j) \in \mathcal{P}_W\}$.

**Assumption C.4** (Off-Band Separation). We assume a margin $\delta > 0$ for the reduced gradient outside the diagonal band:

$$G_{ij}^* \geq \delta > 0, \quad \forall(i, j) \notin \mathcal{P}_W. \tag{43}$$

**Proposition C.5** (Banded Support Containment). *Under the Off-Band Separation assumption, $S^* \subseteq \mathcal{P}_W$, and consequently $|S^*| \leq (2W + 1)n = O(n)$.*

*Proof.* Similar to the $\kappa$-NN case, for any off-band pair $(i, j) \notin \mathcal{P}_W$, Assumption C.4 guarantees $G_{ij}^* \geq \delta > 0$. By strict complementarity, this implies $\pi_{ij}^* = 0$. Thus, the support $S^*$ is confined within the band $\mathcal{P}_W$. With the row-wise bound $2W + 1$, Lemma C.1 yields the linear complexity $|S^*| = O(n)$. □

## D. Convergence Analysis of LoBCD-GW

In this appendix, we establish the global convergence of the LoBCD-GW algorithm. First, Lemma 3.7 confirms the *Sufficient Decrease* property, ensuring a monotonic reduction in the objective value. Based on this foundation, Theorem 3.8 proves that the algorithm asymptotically converges to a stationary point.

Our analysis relies on the following standard assumptions:

1. Smoothness. The GW objective $L(\pi)$, as defined in (3), is $L_g$-smooth with respect to the Frobenius norm.

2. Strong Convexity. The KL divergence $\mathrm{KL}(\cdot \| \pi^{(k)})$ in (5) is $c_{KL}$-strongly convex with respect to the Frobenius norm on the feasible set.

### D.1. Proof of Sufficient Decrease (Lemma 3.7)

**Lemma 3.7.** Let $L_g$ denote the Lipschitz constant of the gradient $\nabla L(\pi)$ in (4), and let $c_{\mathrm{KL}}$ denote the strong convexity modulus of the proximal term $\mathrm{KL}(\cdot \| \pi^{(k)})$ defined in (5). If the step size satisfies $\tau < 2c_{\mathrm{KL}}/L_g$, the iterates generated by LoBCD-GW satisfy:

$$L(\pi^{(k+1)}) - L(\pi^{(k)}) \leq -\underbrace{\left( \frac{c_{\mathrm{KL}}}{\tau} - \frac{L_g}{2} \right)}_{\rho > 0} \|\pi^{(k+1)} - \pi^{(k)}\|_F^2. \tag{44}$$

Consequently, the objective function value $L(\pi)$ is monotonically non-increasing.

*Proof.* By the $L_g$-smoothness of the GW objective $L(\pi)$, the following quadratic upper bound holds for the iterates $\pi^{(k)}$ and $\pi^{(k+1)}$:

$$L(\pi^{(k+1)}) \leq L(\pi^{(k)}) + \langle \nabla L(\pi^{(k)}), \pi^{(k+1)} - \pi^{(k)} \rangle + \frac{L_g}{2} \|\pi^{(k+1)} - \pi^{(k)}\|_F^2. \tag{45}$$

Recall that $\pi^{(k+1)}$ is defined as the minimizer of the KL-regularized subproblem in (5). The first-order optimality condition implies that for any feasible point $\pi \in \Pi(\mu, \nu)$:

$$\langle \nabla L(\pi^{(k)}) + \frac{1}{\tau} \nabla_\pi \mathrm{KL}(\pi^{(k+1)} \| \pi^{(k)}), \pi - \pi^{(k+1)} \rangle \geq 0. \tag{46}$$

Setting $\pi = \pi^{(k)}$ yields:

$$\langle \nabla L(\pi^{(k)}), \pi^{(k+1)} - \pi^{(k)} \rangle \leq \frac{1}{\tau} \langle \nabla_\pi \mathrm{KL}(\pi^{(k+1)} \| \pi^{(k)}), \pi^{(k)} - \pi^{(k+1)} \rangle. \tag{47}$$

Furthermore, since the proximal term $\mathrm{KL}(\cdot \| \pi^{(k)})$ is $c_{\mathrm{KL}}$-strongly convex, its gradient satisfies the strong monotonicity property. Specifically, utilizing $\nabla_\pi \mathrm{KL}(\pi^{(k)} \| \pi^{(k)}) = 0$, we have:

$$\langle \nabla_\pi \mathrm{KL}(\pi^{(k+1)} \| \pi^{(k)}) - \underbrace{\nabla_\pi \mathrm{KL}(\pi^{(k)} \| \pi^{(k)})}_{0}, \pi^{(k+1)} - \pi^{(k)} \rangle \geq c_{\mathrm{KL}} \| \pi^{(k+1)} - \pi^{(k)} \|_F^2. \tag{48}$$

Multiplying by $-1$ reverses the inequality:

$$\langle \nabla_\pi \mathrm{KL}(\pi^{(k+1)} \| \pi^{(k)}), \pi^{(k)} - \pi^{(k+1)} \rangle \leq -c_{\mathrm{KL}} \| \pi^{(k+1)} - \pi^{(k)} \|_F^2. \tag{49}$$

Substituting this result into (47), we obtain a tighter bound on the inner product:

$$\langle \nabla L(\pi^{(k)}), \pi^{(k+1)} - \pi^{(k)} \rangle \leq -\frac{c_{\mathrm{KL}}}{\tau} \| \pi^{(k+1)} - \pi^{(k)} \|_F^2. \tag{50}$$

Finally, substituting this back into the smoothness inequality (45) gives:

$$L(\pi^{(k+1)}) - L(\pi^{(k)}) \leq -\frac{c_{\mathrm{KL}}}{\tau} \| \pi^{(k+1)} - \pi^{(k)} \|_F^2 + \frac{L_g}{2} \| \pi^{(k+1)} - \pi^{(k)} \|_F^2$$

$$= -\left( \frac{c_{\mathrm{KL}}}{\tau} - \frac{L_g}{2} \right) \| \pi^{(k+1)} - \pi^{(k)} \|_F^2, \tag{51}$$

which establishes the sufficient decrease property. $\qquad\square$

### D.2. Proof of Global Convergence (Theorem 3.8)

**Theorem 3.8.** To establish convergence, we utilize the *Global Proximal Gradient Mapping*, defined as:

$$G_\tau(\pi^{(k)}) := \frac{1}{\tau} \left( \pi^{(k)} - \mathrm{Proj}_{\Pi(\mu,\nu)}^{\mathrm{KL}} \left( \pi^{(k)} \odot \exp \left( -\tau \nabla L(\pi^{(k)}) \right) \right) \right). \tag{52}$$

Given the sufficient decrease property established in Lemma 3.7, the global proximal gradient mapping of the iterates generated by LoBCD-GW satisfies:

$$\lim_{k \to \infty} \| G_\tau(\pi^{(k)}) \|_F = 0. \tag{53}$$

This implies that every limit point of the sequence $\{\pi^{(k)}\}$ is a stationary point of the original GW problem.

*Proof.* Step 1: Vanishing Updates.
Summing the inequality from Lemma 3.7 over $k = 0$ to $T$:

$$\sum_{k=0}^{T} \rho \| \pi^{(k+1)} - \pi^{(k)} \|_F^2 \leq L(\pi^{(0)}) - L(\pi^{(T+1)}) \leq L(\pi^{(0)}) - L^* < \infty. \tag{54}$$

The convergence of the series implies that the update magnitude must vanish:

$$\lim_{k \to \infty} \| \pi^{(k+1)} - \pi^{(k)} \|_F = 0. \tag{55}$$

Step 2: From Local to Global.
Since our block selection strategy $\mathcal{A}_k$ (defined in (7)) selects coordinates with the largest projected gradient updates, it inherently satisfies the *Gauss-Southwell efficiency property* (Nutini et al., 2015). Specifically, there exists an efficiency parameter $\eta \in (0, 1]$ such that:

$$\| \pi^{(k+1)} - \pi^{(k)} \|_F \geq \eta \tau \| G_\tau(\pi^{(k)}) \|_F. \tag{56}$$

Rearranging this inequality for $\|G_\tau\|_F$:

$$\|G_\tau(\pi^{(k)})\|_F \leq \frac{1}{\eta\tau}\|\pi^{(k+1)} - \pi^{(k)}\|_F. \tag{57}$$

Taking the limit as $k \to \infty$, and using the result from Step 1, we conclude:

$$\lim_{k\to\infty} \|G_\tau(\pi^{(k)})\|_F = 0. \tag{58}$$

This completes the proof. $\qquad\square$

# E. Additional Experiment Details

*Table 2.* Dataset statistics and descriptions.

| Dataset | Ave. Nodes | Ave. Edges | Description |
|---|---|---|---|
| Douban-230 | 234.00 | 345.33 | Social Network |
| Reddit | 375.90 | 449.30 | Social Network |
| Synthetic | 1500.00 | 56579.00 | Synthetic Network |
| ACM-1000 | 1000.00 | 4163.00 | Co-Author Network |
| DBLP-1000 | 1000.00 | 3747.00 | Co-Author Network |

*Table 3.* Runtimes of LoBCD-GW at different node scales.

| Number of Nodes | Running Time (seconds) |
|---|---|
| 1000 | 0.4658 |
| 2000 | 1.8551 |
| 3000 | 1.9682 |
| 4000 | 3.6038 |
| 5000 | 5.2929 |

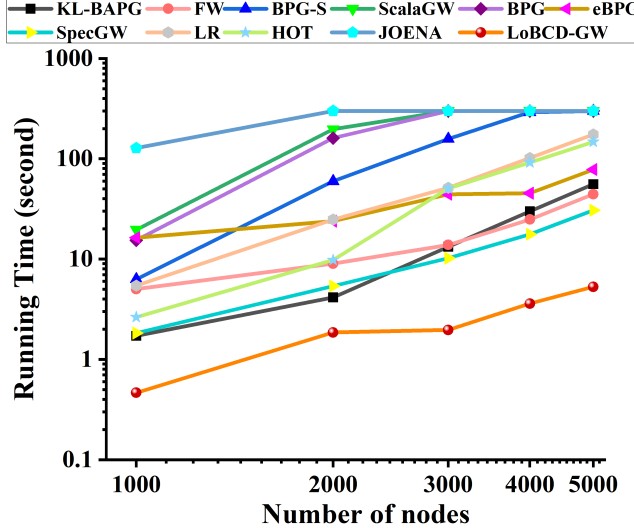

*Figure 5.* Log-scale plot of runtime versus the number of nodes.

## E.1. Ablation Studies

*Table 4.* Comparison of alignment accuracy (%) and wall-clock time (seconds) for graph alignment. Bold indicates the best performance.

| Method | Douban-230 | | Reddit | | Synthetic | | ACM-1000 | | DBLP-1000 | |
|---|---|---|---|---|---|---|---|---|---|---|
| | Acc | Time | Acc | Time | Acc | Time | Acc | Time | Acc | Time |
| LoBCD-GW w/o Local $M$ | **99.72** | 0.90 | **98.94** | 1.23 | 100.0 | 6.19 | **99.20** | 4.95 | 97.30 | 3.65 |
| LoBCD-GW w/o Local Sink | 99.71 | 0.51 | 97.20 | 0.28 | 100.0 | 1.01 | 99.03 | 0.76 | 97.30 | 0.85 |
| LoBCD-GW | 99.71 | **0.45** | 97.18 | **0.22** | **100.0** | **0.27** | 99.03 | **0.49** | **97.30** | **0.52** |

### E.2. The Robustness of LoBCD-GW

In this section, we conduct a series of graph alignment experiments under noisy conditions to further demonstrate the robustness of LoBCD-GW. Specifically, following the protocol in (Xu et al., 2019b), we introduce $q\%$ noisy nodes and $q\%$ noisy edges to the graph, where $q = 10$. We report the alignment accuracy (Mean $\pm$ Std.) averaged over five independent trials with distinct random seeds in the noise generation process. As summarized in Table 5, LoBCD-GW demonstrates superior robustness, achieving the best performance in four out of five datasets while remaining stable across varying random seeds in the noise generation process.

*Table 5.* Robustness assessment of various GW methods under noisy conditions. Results (Mean $\pm$ Std.) are averaged over five independent trials with distinct random seeds for noise generation.

| Method | Douban-230-Noisy | Reddit-Noisy | ACM-1000-Noisy | DBLP-1000-Noisy | Synthetic |
|---|---|---|---|---|---|
| eBPG | 5.46±0.58 | 3.34±0.03 | 0.40±0.17 | 0.10±0.00 | 34.33±0.61 |
| BPG-S | 18.01±0.73 | 36.68±0.02 | 0.93±0.03 | 0.20±0.03 | 61.48±0.34 |
| BPG | 18.44±0.72 | 36.68±0.02 | 0.93±0.06 | 0.20±0.03 | 57.56±0.45 |
| FW | 31.30±0.98 | 18.34±0.22 | 88.30±0.97 | 71.87±0.83 | 24.50±0.95 |
| KL-BAPG | 67.35±0.56 | 49.45±0.02 | 91.07±0.43 | 86.13±0.57 | 99.79±0.21 |
| JOENA | 72.31±0.53 | **85.36±0.57** | 87.04±0.39 | 87.25±0.21 | 82.26±0.53 |
| ScalaGW | 2.30±0.17 | 0.70±0.05 | 1.08±0.09 | 1.36±0.40 | 17.93±0.27 |
| SpecGW | 31.76±0.96 | 19.66±0.16 | 28.50±0.95 | 14.67±0.80 | 13.27±0.49 |
| LR | 61.28±0.42 | 1.26±0.03 | 4.37±0.09 | 1.60±0.04 | 7.24±0.19 |
| HOT | 39.90±0.87 | **69.29±0.26** | 87.48±0.51 | 56.28±0.37 | 100.0±0.00 |
| LoBCD-GW | **78.42±0.89** | 62.23±0.26 | **92.27±0.40** | **87.67±0.03** | **100.0±0.00** |

### E.3. Additional Comparisons on Graph Alignment

This section reports additional graph-alignment comparisons with FAQ (Vogelstein et al., 2015), Sampled-GW (Kerdoncuff et al., 2021), SparGW (Li et al., 2023b), FGC-GW (Zhang et al., 2024a), CSGO (Shen et al., 2026), ASM (Shen et al., 2024), and FRAM (Shen & Zhu, 2026) to further assess LoBCD-GW against a broader set of representative baselines. Table 6 shows that LoBCD-GW achieves the best or tied-best accuracy on four out of five datasets. On the three large-scale datasets, LoBCD-GW consistently attains the fastest runtime while maintaining the best or tied-best accuracy, demonstrating the effectiveness of our method.

*Table 6.* Additional comparisons of alignment accuracy (%) and wall-clock time (seconds) across small-scale (Douban-230, Reddit) and large-scale (Synthetic, ACM-1000, DBLP-1000) datasets. Bold indicates the best performance. All methods are evaluated on CPU for fair comparison.

| | Douban-230 | | Reddit | | Synthetic | | ACM-1000 | | DBLP-1000 | |
|---|---|---|---|---|---|---|---|---|---|---|
| Method | Acc | Time | Acc | Time | Acc | Time | Acc | Time | Acc | Time |
| FAQ | 85.33 | 1.81 | 40.20 | 4.99 | 64.62 | 42.96 | 92.53 | 12.27 | 79.83 | 20.99 |
| Sampled-GW | 21.62 | 22.52 | 36.87 | 34.91 | 38.16 | 160.30 | 44.50 | 147.65 | 6.87 | 126.85 |
| SparGW | **100.0** | 10.29 | 95.61 | 26.97 | **100.0** | 46.50 | 97.40 | 72.24 | 94.27 | 87.14 |
| FGC-GW | 86.13 | 5.04 | 79.72 | 8.97 | **100.0** | 22.85 | 94.27 | 25.50 | 87.37 | 11.94 |
| CSGO | 52.03 | **0.10** | 73.38 | 0.42 | 63.40 | 29.36 | 91.60 | 2.33 | 15.93 | 2.30 |
| ASM | 90.77 | 0.60 | 43.02 | 0.43 | **100.0** | 9.85 | 98.07 | 8.23 | 95.17 | 4.87 |
| FRAM | 96.01 | 0.80 | 83.30 | **0.03** | **100.0** | 1.50 | 96.43 | 1.20 | 94.13 | 1.30 |
| LoBCD-GW (CPU) | 99.71 | 0.52 | **97.18** | 0.57 | **100.0** | **1.14** | **99.03** | **0.63** | **97.30** | **0.65** |

### E.4. Feasibility Analysis and Gradient Analysis

In this section, we conduct a comprehensive feasibility analysis through extensive experiments. As shown in Table 7, FW and SpecGW leverage Linear Programming (LP) as the inner solver, thereby yielding near-zero marginal errors. Similarly,

our proposed LoBCD-GW, alongside eBPG and BPG, employs the Sinkhorn solver, which also results in negligible marginal errors. This demonstrates the effectiveness of our marginal compensation mechanism, ensuring strict feasibility by synchronizing local mass redistribution with global constraints. In contrast, methods based on constraint relaxation, such as BPG-S and KL-BAPG, tend to exhibit inherent marginal errors. In particular, ScalaGW introduces global marginal errors because its recursive hard partitioning mechanism restricts the feasible set of the coupling matrix.

We also conduct gradient analysis experiments of LoBCD-GW with respect to the selection threshold $\epsilon_\Delta$. As shown in Table 8, we report the average error and the cosine similarity between the local gradient $\nabla \tilde{L}(\pi)$ and the global gradient $\nabla L(\pi)$ under various selection thresholds $\epsilon_\Delta \in \{10^{-8}, 10^{-7}, 2.5 \times 10^{-7}, 5 \times 10^{-7}, 2.5 \times 10^{-6}\}$. The gradient errors, ranging from $10^{-5}$ to $10^{-3}$ across all datasets, represent a negligible discrepancy. This demonstrates that our selection strategy in Section 3.1, which eliminates redundant computations for stable entries, significantly enhances efficiency with minimal impact on accuracy. Furthermore, the cosine similarity between global and local gradients remains between 0.99 and 1.0, indicating that our local updates maintain high fidelity to the global descent direction. To further demonstrate that the small gradient error has a negligible impact on the graph alignment task, we incorporated a global gradient computation every 50 iterations during the optimization process to further minimize the gradient error. Notably, this periodic refinement resulted in virtually no change to the matching accuracy.

*Table 7.* Comparison of the average marginal error $\|\pi^\top \mathbf{1}_n - \nu\| + \|\pi \mathbf{1}_m - \mu\|$ across five graph alignment datasets.

| Method | Douban-230 | | | Reddit | | | Synthetic | | | ACM-1000 | | | DBLP-1000 | | |
|---|---|---|---|---|---|---|---|---|---|---|---|---|---|---|---|
| | Acc | Time | Error | Acc | Time | Error | Acc | Time | Error | Acc | Time | Error | Acc | Time | Error |
| FW | 34.00 | 0.11 | <1e-10 | 21.51 | 6.31 | <1e-10 | 24.50 | 27.29 | 4.6e-9 | 88.90 | 80.87 | 1.5e-9 | 74.53 | 122.36 | 3.1e-9 |
| SpecGW | 66.08 | 0.03 | <1e-10 | 50.71 | 0.19 | <1e-10 | 13.27 | 4.23 | <1e-10 | 93.73 | 0.50 | <1e-10 | 86.40 | 0.53 | <1e-10 |
| LoBCD-GW | 99.71 | 0.45 | 2.6e-7 | 97.18 | 0.22 | 7.1e-6 | 100.0 | 0.27 | 6.1e-8 | 99.03 | 0.49 | 4.2e-9 | 97.30 | 0.52 | 4.2e-9 |
| eBPG | 6.27 | 1.65 | 2.7e-8 | 3.76 | 1.27 | 2.9e-6 | 34.33 | 13.70 | 3.0e-10 | 0.43 | 5.49 | 2.1e-8 | 0.20 | 1.91 | 2.4e-8 |
| BPG | 24.44 | 0.24 | 2.5e-6 | 39.04 | 1.07 | 1.3e-6 | 57.56 | 31.44 | 2.5e-7 | 1.37 | 4.58 | 6.9e-8 | 0.30 | 4.17 | 1.0e-10 |
| BPG-S | 24.43 | 0.25 | 4.1e-6 | 39.04 | 0.98 | 7.8e-6 | 61.48 | 30.55 | 9.9e-6 | 1.37 | 4.59 | 6.5e-8 | 0.30 | 4.09 | 7.5e-9 |
| KL-BAPG | 68.19 | 0.66 | 1.4e-7 | 50.93 | 0.23 | 5.1e-6 | 99.79 | 4.18 | 2.9e-3 | 94.00 | 1.46 | 1.5e-7 | 86.40 | 1.42 | 1.8e-07 |
| ScalaGW | 2.53 | 0.30 | 7.4e-2 | 0.54 | 0.41 | 1.9e-1 | 17.93 | 24.53 | 1.4e-2 | 1.17 | 4.82 | 2.9e-2 | 1.18 | 3.55 | 2.6e-2 |

*Table 8.* Comparison of accuracy (Acc), gradient error (Error, Error $= \|\nabla L(\pi) - \nabla \tilde{L}(\pi)\|_F$), and local-global gradient cosine similarity (CosSim, CosSim $= \langle \nabla L(\pi), \nabla \tilde{L}(\pi) \rangle / (\|\nabla L(\pi)\|_F \|\nabla \tilde{L}(\pi)\|_F)$) on five datasets. * denotes that LoBCD-GW is executed with a global gradient computation performed every 50 epochs.

| LoBCD-GW | Douban-230 | | | Reddit | | | Synthetic | | | ACM-1000 | | | DBLP-1000 | | |
|---|---|---|---|---|---|---|---|---|---|---|---|---|---|---|---|
| | Acc | Error | CosSim | Acc | Error | CosSim | Acc | Error | CosSim | Acc | Error | CosSim | Acc | Error | CosSim |
| $\epsilon_\Delta = 10^{-8}$ | 99.42 | 2.3e-3 | 0.99 | 99.63 | 3.0e-4 | 1.00 | 100 | 7.5e-5 | 1.00 | 98.90 | 5.4e-4 | 1.00 | 96.27 | 9.3e-4 | 0.99 |
| $\epsilon_\Delta = 10^{-8*}$ | 99.42 | 1.7e-3 | 0.99 | 99.69 | 1.4e-4 | 1.00 | 100 | 3.6e-5 | 1.00 | 98.90 | 3.1e-4 | 1.00 | 96.30 | 2.8e-4 | 1.00 |
| $\epsilon_\Delta = 10^{-7}$ | 99.42 | 4.1e-3 | 0.99 | 98.65 | 5.6e-4 | 1.00 | 100 | 9.2e-6 | 1.00 | 98.70 | 9.3e-4 | 0.99 | 96.29 | 6.2e-4 | 0.99 |
| $\epsilon_\Delta = 10^{-7*}$ | 99.42 | 3.4e-3 | 0.99 | 99.23 | 4.4e-4 | 1.00 | 100 | 2.8e-6 | 1.00 | 98.90 | 3.7e-4 | 0.99 | 96.30 | 2.1e-4 | 0.99 |
| $\epsilon_\Delta = 2.5 \times 10^{-7}$ | 99.10 | 6.8e-3 | 0.99 | 98.62 | 5.6e-4 | 1.00 | 100 | 4.1e-5 | 1.00 | 89.96 | 9.3e-4 | 0.99 | 96.30 | 8.3e-4 | 0.99 |
| $\epsilon_\Delta = 2.5 \times 10^{-7*}$ | 99.13 | 5.4e-3 | 0.99 | 98.83 | 1.8e-4 | 1.00 | 100 | 2.1e-5 | 1.00 | 99.03 | 6.0e-4 | 0.99 | 96.30 | 1.0e-4 | 1.00 |
| $\epsilon_\Delta = 5 \times 10^{-7}$ | 99.71 | 2.5e-3 | 0.99 | 98.20 | 7.0e-4 | 1.00 | 100 | 3.3e-5 | 1.00 | 90.72 | 5.4e-4 | 0.99 | 90.27 | 6.8e-4 | 0.99 |
| $\epsilon_\Delta = 5 \times 10^{-7*}$ | 99.71 | 2.0e-3 | 0.99 | 98.22 | 3.8e-4 | 1.00 | 100 | 8.0e-6 | 1.00 | 90.77 | 1.3e-4 | 0.99 | 90.27 | 2.6e-4 | 0.99 |
| $\epsilon_\Delta = 2.5 \times 10^{-6}$ | 99.68 | 7.3e-3 | 0.99 | 96.97 | 6.2e-4 | 1.00 | 100 | 7.3e-5 | 1.00 | 88.69 | 9.8e-4 | 0.98 | 83.91 | 5.3e-4 | 0.99 |
| $\epsilon_\Delta = 2.5 \times 10^{-6*}$ | 99.71 | 1.2e-3 | 0.99 | 97.18 | 2.8e-4 | 1.00 | 100 | 3.2e-5 | 1.00 | 89.03 | 4.6e-4 | 0.99 | 83.93 | 1.9e-4 | 0.99 |

## E.5. GW Objective Value Analysis

To further examine the optimization behavior of different methods beyond downstream task accuracy, we conduct experiments on two 12-node datasets constructed following the SDP (Chen et al., 2024) setting: a Gaussian point cloud dataset and graphs generated from the Stochastic Block Model (SBM).

The Gaussian point cloud dataset evaluates GW alignment in a continuous geometric setting: we sample 12 points from a Gaussian distribution, use pairwise Euclidean distances as the distance matrix, and perform self-alignment. The SBM-generated graphs are used to evaluate performance in a setting prone to multiple local minima: we partition 12 nodes into two communities with intra- and inter-community edge probabilities 0.7 and 0.1, respectively, and use the adjacency matrix for self-alignment.

On the Gaussian point cloud dataset, LoBCD-GW achieves a GW loss of $2.96 \times 10^{-5}$, which is close to the zero loss

attained by FW and SDP. This suggests that, on this small continuous instance, our method attains a near-globally optimal GW objective value. BAPG and BPG-S obtain slightly higher losses, while eBPG performs substantially worse, likely because its additional regularization shifts the optimization away from the original GW objective.

On the SBM-generated graphs, the symmetry of the community structure makes the problem more susceptible to multiple local minima. Despite this, LoBCD-GW achieves a GW loss close to that of SDP and lower than those of the other baselines. This suggests that the proposed localized block selection strategy focuses optimization on the most critical coupling entries while preserving feasibility, which may help reduce the risk of poor local minima in practice.

*Table 9.* GW objective values on the Gaussian point cloud dataset and SBM-generated graphs.

| Method | Gaussian Point Cloud | SBM-generated Graphs |
|---|---|---|
| SDP | 0.00 | 0.029 |
| FW | 0.00 | 0.054 |
| LoBCD-GW | $2.96 \times 10^{-5}$ | 0.033 |
| BAPG | $6.10 \times 10^{-5}$ | 0.056 |
| BPG-S | $1.09 \times 10^{-4}$ | 0.056 |
| eBPG | $3.94 \times 10^{-2}$ | 0.139 |

### E.6. Graph Partitioning

The GW distance can also be potentially applied to the graph partitioning task. That is, we are trying to match the source graph with a disconnected target graph having $Q$ isolated and self-connected super-nodes, where $Q$ is the number of clusters.

Graph partitioning aims to divide the nodes in a graph into $Q$ clusters such that intra-cluster connections are dense while inter-cluster connections are sparse. Besides the applications to graph alignment, recent works (Abrishami et al., 2020; Chowdhury & Needham, 2021) have illuminated the potential of GW for the graph partitioning problem. Following their formulation, we formalize this task as a GW alignment problem between the source graph and a target cluster prototype graph. This target graph comprises $Q$ isolated super-nodes with self-loops. We compare LoBCD-GW against a range of baselines: (1) the GW-based methods detailed in Section 4.1; (2) three widely used graph clustering methods: FastGreedy (Clauset et al., 2004), Louvain (Blondel et al., 2008), and Infomap (Rosvall & Bergstrom, 2008). These methods partition nodes into clusters based on edge density or information flow, without relying on optimal transport.

*Table 10.* Comparison of AMI scores on graph partitioning datasets. Bold indicates the best performance.

| Category | Method | Wikipedia | | EU-email | | Amazon | | Village | |
|---|---|---|---|---|---|---|---|---|---|
| | | Raw | Noisy | Raw | Noisy | Raw | Noisy | Raw | Noisy |
| Non-GW | FastGreedy | 0.382 | 0.341 | 0.312 | 0.251 | 0.637 | **0.573** | **0.881** | **0.778** |
| | Louvain | 0.377 | 0.329 | 0.447 | 0.382 | 0.622 | **0.584** | **0.881** | **0.827** |
| | Infomap | 0.332 | 0.329 | 0.374 | 0.379 | **0.940** | 0.463 | **0.881** | 0.190 |
| GW | eBPG | 0.100 | 0.082 | 0.011 | 0.188 | 0.604 | 0.031 | 0.002 | 0.003 |
| | FW | 0.334 | 0.299 | 0.441 | 0.424 | 0.337 | 0.326 | 0.640 | 0.515 |
| | BPG-S | 0.411 | 0.373 | 0.475 | 0.253 | 0.483 | 0.425 | 0.642 | 0.619 |
| | BPG | 0.418 | 0.373 | 0.473 | 0.253 | 0.492 | 0.436 | 0.705 | 0.619 |
| | LR | 0.423 | 0.345 | 0.026 | 0.002 | 0.632 | 0.302 | 0.760 | 0.013 |
| | SpecGW | 0.442 | 0.395 | 0.487 | 0.425 | 0.565 | 0.487 | 0.758 | 0.707 |
| | KL-BAPG | **0.533** | 0.365 | 0.533 | 0.436 | 0.630 | 0.502 | 0.797 | 0.711 |
| | LoBCD-GW | 0.518 | **0.442** | **0.554** | **0.453** | **0.646** | **0.520** | **0.808** | **0.751** |

**Parameter Setup.** For the input distance matrices, we employ heat kernel matrices (Chowdhury & Needham, 2021) to capture the multi-scale topological structure. The heat kernel is defined as $H_t = \exp(-tS)$, where $S$ denotes the normalized graph Laplacian. The diffusion time $t$ is set consistently with the configuration in (Chowdhury & Needham, 2021). For LoBCD-GW, the step size $\tau$ is selected from $[10, 1000]$, and the selection threshold $\epsilon_\Delta$ is selected from

$\{10^{-8}, 10^{-7}, 2.5 \times 10^{-7}, 5 \times 10^{-7}, 2.5 \times 10^{-6}\}$. We report the best results via the Adjusted Mutual Information (AMI) score (Vinh et al., 2009), where higher values indicate better clustering consistency.

**Results and Discussion.** Table 10 reports the AMI of all the methods on four benchmark datasets (Chowdhury & Needham, 2021), evaluated using both the raw (Raw) and noisy (Noisy) versions. Compared with GW baselines, LoBCD-GW achieves the best performance on the vast majority of datasets and consistently attains the highest AMI across all Noisy settings, demonstrating strong robustness. Compared with non-GW baselines, LoBCD-GW also shows strong competitiveness, achieving 5, 5, and 6 best scores out of 8 Raw/Noisy evaluations when compared against FastGreedy, Louvain, and Infomap, respectively. Nevertheless, LoBCD-GW is not always superior to non-GW methods in several Raw settings, especially on Amazon and Village. This gap may mainly stem from limitations of the GW formulation for partitioning tasks. First, modeling partitioning as alignment to a disconnected prototype assumes an idealized topology and ignores the inter-community edges prevalent in complex networks such as Amazon and Village. Second, GW enforces strict mass conservation, which may conflict with highly imbalanced community sizes and thus cause unnatural splitting or merging. In contrast, non-GW methods directly optimize modularity or information flow, making them better suited to such topologies.

### E.7. Multi-omics Single-cell Integration

To further assess the generalization of LoBCD-GW beyond graph alignment, we extend it to the frontier task of single-cell multi-omics data integration (Miao et al., 2021; Hu et al., 2024). The goal is to jointly model cross-modal molecular representations at single-cell resolution, for example gene expression with DNA methylation (Teichmann & Efremova, 2020; Wen et al., 2022; Cao et al., 2020), to obtain a more complete and unified characterization of cellular states. However, this problem is intrinsically challenging because multi-omics datasets typically contain unpaired cells across modalities, exhibit mismatched feature spaces, and present pronounced nonlinear geometric distortions, which together make robust integration highly nontrivial.

**Task Definition.** Let $E = [E_1, \ldots, E_{n_e}]^\top \in \mathbb{R}^{n_e \times d_e}$ and $M = [M_1, \ldots, M_{n_m}]^\top \in \mathbb{R}^{n_m \times d_m}$ denote two single-cell multi-omics datasets across distinct modalities, where $n_e$ and $n_m$ represent the number of cells in the two modalities respectively, while $d_e$ and $d_m$ denote their corresponding feature dimensions. The goal of single-cell multi-omics integration is to align samples belonging to the same cell type across different modalities.

*Table 11.* Statistics of real-world single-cell multi-omics datasets.

| Modality Name | scGEM | | scNMT | | |
|---|---|---|---|---|---|
| | Gene Expression | DNA Methylation | Gene Expression | DNA Methylation | Chromatin Accessibility |
| Samples $n$ | 177 | 177 | 612 | 709 | 1773 |
| Features $d$ | 34 | 27 | 300 | 300 | 300 |

**Dataset and Evaluation.** We validated our method on two real-world single-cell multi-omics datasets: scGEM (gene expression and DNA methylation) and scNMT (gene expression, DNA methylation, and chromatin accessibility). Dataset statistics are summarized in Table 11. To evaluate our method, we perform a comprehensive comparison of the label transfer accuracy (%) and wall-clock time (seconds). Specifically, for alignment quality, we employ cross-modal label transfer accuracy by transferring cell type labels across modalities and calculating the proportion of predictions consistent with ground-truth annotations, reporting these results as percentages (%). Additionally, to assess computational efficiency, we report the end-to-end wall-clock time in seconds.

**Baselines and Parameter Setup.** We evaluate LoBCD-GW against the GW methods listed in Section 4.1. Specifically, we construct the input distance matrices using cosine similarity computed from features reduced via Principal Component Analysis (PCA) (Abdi & Williams, 2010). We perform a grid search over hyperparameters, with the number of PCA components in $\{2, 4, 6, 14\}$ and the step size $\tau$ in $\{10^2, 10^3, 10^4, 10^5\}$.

**Results of All Methods.** As shown in Table 12, LoBCD-GW achieves the best performance in most label transfer tasks and consistently attains the shortest running time. Notably, when compared with two strong GW baselines, KL-BAPG and ScalaGW, LoBCD-GW yields substantial improvements: on the scGEM dataset, it improves accuracy by 6.8–7.4 percentage points with a $1.8\times$–$4.1\times$ speedup; on the scNMT dataset, it achieves 0.3–6.5 percentage-point accuracy gains on five out of six transfer directions, with a $2.6\times$–$58.1\times$ speedup. These results demonstrate the potential of LoBCD-GW

*Table 12.* Comparison of the label transfer accuracy (%) and wall-clock time (seconds) on scGEM and scNMT for single-cell multi-omics integration. $D.$, $G.$, and $Ch.$ denote DNA methylation, gene expression, and chromatin accessibility, respectively. Bold indicates the best performance.

| | scGEM | | | scNMT | | | | | | |
|---|---|---|---|---|---|---|---|---|---|---|
| Method | D.→G. | G.→D. | Time | D.→G. | G.→D. | Ch.→G. | G.→Ch. | D.→Ch. | Ch.→D. | Time |
| eBPG | 7.9 | 10.7 | 7.4 | 63.4 | 68.0 | 46.6 | 44.8 | 54.2 | 38.6 | 92.38 |
| BPG-S | 9.0 | 14.1 | 1.2 | 69.0 | 73.2 | 60.1 | 60.7 | 25.2 | 25.6 | 24.98 |
| BPG | 11.9 | 11.3 | 11.5 | 75.1 | 73.2 | 60.8 | 60.0 | 25.0 | 25.6 | 40.50 |
| SpecGW | 21.5 | 21.5 | 0.4 | 61.4 | 57.2 | 36.4 | 39.7 | 38.2 | 39.3 | 8.02 |
| FW | 40.7 | 40.7 | 0.7 | 74.2 | 74.3 | 44.1 | 45.4 | 38.5 | 39.3 | 36.39 |
| LR | 50.8 | 52.0 | 0.34 | 61.1 | 62.1 | 68.1 | 69.3 | 60.3 | 58.4 | 16.29 |
| KL-BAPG | 52.0 | 50.8 | 0.3 | 74.0 | 74.3 | 68.0 | 67.0 | 39.4 | 34.5 | 1.78 |
| ScalaGW | 53.1 | 53.1 | 0.7 | 74.3 | 73.5 | 69.5 | 67.2 | **82.2** | 38.6 | 40.10 |
| LoBCD-GW | **60.5** | **59.9** | **0.17** | **75.5** | **77.3** | **69.8** | **73.7** | 73.5 | **42.5** | **0.69** |

for broader applications, highlighting its superior scalability, which is crucial for handling large-scale single-cell datasets.

### E.8. Source code of the baselines used in this paper and their licenses

- **BPG-S** (GPL-3.0) (Xu et al., 2019b): `https://github.com/HongtengXu/gwl`

- **ScalaGW** (No license) (Xu et al., 2019a): `https://github.com/HongtengXu/s-gwl`

- **SpecGW** (MIT) (Chowdhury & Needham, 2021): `https://github.com/trneedham/Spectral-Gromov-Wasserstein`

- **eBPG** and **FW** (MIT) (Flamary et al., 2021): `https://github.com/PythonOT/POT`

- **IPFP**, **RRWM**, **SpecMethod** (Mulan PSL v2): `https://github.com/Thinklab-SJTU/pygmtools`

- **LR** (Scetbon et al., 2022): `https://github.com/meyerscetbon/LinearGromov`

- **KL-BAPG** (Li et al., 2023a): `https://github.com/squareRoot3/Gromov-Wasserstein-for-Graph`

- **HOT** (Zeng et al., 2024): `https://github.com/zhichenz98/HOT-AAAI24`

- **JOENA** (Yu et al., 2025): `https://github.com/yq-leo/JOENA-WWW25`

- **FAQ** (Vogelstein et al., 2015): `https://github.com/jovo/FastApproximateQAP`

- **Sampled-GW** (Kerdoncuff et al., 2021): `https://github.com/Hv0nnus/Sampled-Gromov-Wasserstein`

- **SparGW** (Li et al., 2023b): `https://github.com/Mengyu8042/Spar-GW`

- **CSGO** (Shen et al., 2026): `https://github.com/BinruiShen/CSGO`

- **ASM** (Shen et al., 2024): `https://github.com/BinruiShen/Adaptive-Softassign-Matching`

- **FRAM** (Shen & Zhu, 2026): `https://github.com/BinruiShen/FRAM`

