# OpenReview forum: "LoBCD-GW: A Fast and Data-Dependent Algorithm for Computing Gromov-Wasserstein Distance via Localized Block Coordinate Descent"
_ICML.cc/2026/Conference — ICML 2026 regular_

### Official Review · Reviewer_PEtp · 2026-03-05

**Soundness:** 3
**Presentation:** 3
**Significance:** 4
**Originality:** 3
**Overall Recommendation:** 5
**Confidence:** 4

**Summary:**

The paper proposes a faster solver for GW based on a localized block coordinate descent strategy. The main idea is to exploit the observation that, during optimization, only a subset of coupling entries changes significantly at each iteration. Instead of updating the full transport plan, the method identifies an active set and constructs a corresponding block, and performs a Sinkhorn-style update only on this subproblem. The paper provides theoretical analysis for the proposed approach, including results on data-dependent sparsity of the selected active set, sufficient decrease, and convergence to a stationary point under stated assumptions. The experiments show that the proposed method achieves runtime reductions while maintaining competitive task performance.

**Compliance With Llm Reviewing Policy:**

Affirmed.

**Final Justification:**

My concerns are resolved. I keep my recommendation for acceptance.

**Key Questions For Authors:**

1. It is quite surprisingly to see from Table 1 that the accuracy of the proposed method is nearly 1. Does it mean the proposed method find the nearly global minimum of the GW distance? Can the author compare with methods with global guarantee (e.g., the SDP relaxation [1]) on some small datasets to see if this is true? If this is true, can the author comment on which part of the algorithm helps to escape from local min compared with other local methods?

2. Theorem 3.1 says the size of the set $\mathcal{A}_k$ become $O(n)$ for sufficiently large $k > K$. But it is unclear how large $K$ should be. Can the author plot the size of $\mathcal{A}_k$ over iterations to see how it decreases?

3. Do the authors use the same initial solution across different methods?

4. Can you also compare the optimal value obtained by different methods? This is to exclude the case where the supports of the obtained solutions are different (which leads to different accuracy) but they correspond to a similar GW distance.

[1] Chen, J., Nguyen, B. T., Koh, S., & Soh, Y. S. (2024). Semidefinite relaxations of the Gromov-Wasserstein distance. Advances in Neural Information Processing Systems, 37, 69814-69839.

**Limitations:**

Limitations are not discussed. Based on my understanding of gradient-based method for nonconvex problem, I think the limitation is that the global optimality can not be guarantee and the optimality gap remains unclear.

**Strengths And Weaknesses:**

Strengths.

1. The paper addresses the high runtime cost of sink-horn based GW solver and proposes a localized update strategy that is well motivated for large-scale problems. The method is conceptually clear.

2. The paper also includes theory (data-dependent sparsity, sufficient decrease, convergence to stationarity under assumptions) that strengthens the technical foundation of the method.

3. The method is evaluated on multiple tasks (graph alignment, graph partitioning, single-cell integration), so the paper demonstrates breadth beyond a single benchmark.

4. The paper is well-written.

Weaknesses.

1. The paper reports high task accuracy (e.g., near-perfect accuracy in Table 1), but it is unclear whether this corresponds to finding near-global minima of the GW objective or simply good downstream solutions under the chosen metric.

2. Relatedly, the paper does not compare final GW objective values across methods, which makes it difficult to assess whether methods with different supports/accuracies may still achieve similar GW losses. A comparison on small instances with methods that provide stronger global guarantees (e.g., SDP-based baselines) would substantially strengthen the paper’s claims and clarify whether the proposed localization also helps avoid poor local minima.

2. Theorem 3.1 states that the size of the set $\mathcal{A}_k$ become $O(n)$ for sufficiently large $k > K$. But it is unclear when does the claimed complexity reduction becomes active in practice and how sensitive it is to problem difficulty.

3. It is not clear whether all methods use the same initialization or how initialization is set.

---

> ### Author Rebuttal · Authors · 2026-03-31
>
> We thank the reviewer for the insightful comments and address the concerns below.
>
> **W1/W2/Q1/Q4** : Regarding GW Objective (Loss)
>
> To address questions related to the GW objective value, we will add experiments on two 12-node datasets constructed following SDP [1]: a Gaussian point cloud dataset and graphs generated from the Stochastic Block Model (SBM). We will **compare our method with SDP [1] and other baselines, and report the final GW objective values** attained by all methods.
>
> **The Gaussian point cloud dataset evaluates GW alignment in a continuous geometric setting**: we sample 12 points from a Gaussian distribution, use pairwise Euclidean distances as the distance matrix, and perform self-alignment. **The SBM-generated graphs are used to evaluate performance in a setting prone to multiple local minima:** we partition 12 nodes into two communities with intra- and inter-community edge probabilities 0.7 and 0.1, respectively, and use the adjacency matrix for self-alignment.
>
> On the Gaussian point cloud dataset, LoBCD-GW achieves a GW loss of $2.96 \times 10^{-5}$, which is close to the zero loss attained by FW and SDP. **This suggests that, on this small continuous instance, our method attains a near-globally optimal GW objective value.** BAPG and BPG-S obtain slightly higher losses, while eBPG performs substantially worse, likely because its additional regularization shifts the optimization away from the original GW objective.
>
> On the SBM-generated graphs, the symmetry of the community structure makes the problem more susceptible to multiple local minima. Despite this, **LoBCD-GW achieves a GW loss close to that of SDP and lower than those of the other baselines**. This suggests that **the proposed localized block selection strategy focuses optimization on the most critical coupling entries while preserving feasibility, which may help reduce the risk of poor local minima in practice.**
>
> Table 1. GW objective values on the Gaussian point cloud dataset
>
> | Method   | GW Loss               |
> | :------- | :-------------------- |
> | SDP      | 0.00                  |
> | FW       | 0.00                  |
> | LoBCD-GW | $2.96 \times 10^{-5}$ |
> | BAPG     | $6.10 \times 10^{-5}$ |
> | BPG-S    | $1.09 \times 10^{-4}$ |
> | eBPG     | $3.94 \times 10^{-2}$ |
>
> Table 2. GW objective values on SBM-generated graphs
>
> | Method   | GW Loss |
> | :------- | :------ |
> | SDP      | 0.029   |
> | LoBCD-GW | 0.033   |
> | FW       | 0.054   |
> | BAPG     | 0.056   |
> | BPG-S    | 0.056   |
> | eBPG     | 0.139   |
>
> **W3/Q2:** Theorem 3.1 states that the size of the set $\mathcal{A}_k$ ... in practice and how sensitive it is to problem difficulty.
>
> We thank the reviewer for this constructive suggestion. **Figure 1 in the original manuscript already shows the evolution of $|\mathcal{A}_k|$ over iterations**. We **have updated it to include the first 50** **iterations** to better illustrate its early-stage behavior, and the updated figure is available at the anonymous link https://anonymous.4open.science/r/r-ED77/Fig1.png. The updated plot shows that **$|\mathcal{A}_k|$ drops sharply in the early iterations, suggesting that the effective $K$ is small in practice and that the claimed $O(n)$ complexity reduction becomes active quickly**. Moreover, on the Reddit dataset (500 graph pairs), the density $\alpha = |\mathcal{A}_k|/mn$ **exhibits a narrow standard deviation, indicating that the shrinkage pattern of $\mathcal{A}_k$ is empirically stable across these instances**.
>
> **W4/Q3:** It is not clear whether all methods use...or how initialization is set.
>
> **The same initialization protocol is used across all evaluated methods to ensure a fair comparison.** For each method, we report the better result obtained from the following two initialization strategies:
>
> - **Standard Independent Coupling:** $\pi^{(0)} \leftarrow \mu \nu^\top$.
>
> - **Degree-based Heuristic Initialization:**
>
> $$
> \pi^{(0)} \leftarrow \operatorname{Proj}_{\Pi(\mu,\nu)}^{\mathrm{KL}}
> \left(
> \exp\left(
> -\frac{|D_s\mathbf{1}-(D_t\mathbf{1})^\top|}
> {\left(\frac{1}{mn}\|D_s\mathbf{1}-(D_t\mathbf{1})^\top\|_1\right)\cdot \mathrm{temp}+\mathrm{eps}}
> \right)
> \right)
> $$
>
> We use temp = 0.1 and eps = 1e-10 in all experiments.
>
> We will also revise Section 4.1 (Parameter Setup) to explicitly specify these initialization protocols in the revised manuscript.

---

> > ### Author Rebuttal · Reviewer_PEtp · 2026-04-02
> >
> > Thank the authors for their detailed reply and additional experiments. The comparison of the GW objective is convincing. I keep my recommendation for acceptance.

---

> > > ### Author Response · Authors · 2026-04-02
> > >
> > > Thank you again for your thoughtful comments. We sincerely appreciate your time and valuable feedback.

---

### Official Review · Reviewer_V5Wi · 2026-03-09

**Soundness:** 2
**Presentation:** 3
**Significance:** 3
**Originality:** 3
**Overall Recommendation:** 4
**Confidence:** 5

**Summary:**

This paper proposes LoBCD-GW, an algorithm for efficiently computing the Gromov–Wasserstein (GW) distance using a localized block coordinate descent strategy. The core idea is to exploit data-dependent sparsity in the coupling updates, restricting computation to a dynamically selected subset of coordinates whose gradients indicate significant updates.  Theoretical results establish convergence guarantees and bounds on the sparsity of active coordinates. Extensive experiments on graph alignment benchmarks demonstrate substantial speedups and strong accuracy compared with prior GW solvers.

**Compliance With Llm Reviewing Policy:**

Affirmed.

**Final Justification:**

This paper proposes LoBCD-GW, an algorithm for efficiently computing the Gromov–Wasserstein (GW) distance via a localized block coordinate descent strategy. Although the practical complexity remains $\mathcal{O}(n^3)$, rather than the claimed $\mathcal{O}(r^3)$, the method still demonstrates impressive performance at the current problem scale. On the other hand, the GPU acceleration is relatively limited (around 50% on average), suggesting that the algorithm is not particularly GPU-friendly—possibly due to communication overhead caused by a large number of iterations. Overall, I lean toward a **weak accept**.

**Key Questions For Authors:**

1. The block selection rule is defined using the full projected update$\Delta\pi^{(k)}$. Does constructing $A_k​$ require evaluating the full gradient and a full KL projection, at least in the first iteration?

2. Could the authors provide the exact runtime values of LoBCD-GW at each scale in Figure 4? Presenting the concrete running times would make it easier for readers to perform a direct comparison across different problem sizes.

**Limitations:**

yes

**Strengths And Weaknesses:**

### Strength

1. **Interesting algorithmic insight**
The key observation behind the proposed method is that **most entries in the coupling matrix quickly stabilize during optimization**, while only a small subset undergoes significant updates. By restricting computation to this dynamically selected subset, the authors reduce the per-iteration cost from $O(n^3)$ to $O(r^3)$, where $r \ll n$.

2. **Feasibility-preserving update mechanism**
Localized updates often break global transport constraints. The proposed marginal compensation mechanism addresses this issue by redistributing mass to maintain feasibility. This is an elegant design that allows localized optimization while still satisfying the marginal constraints.

3. **Theoretical analysis**
The paper provides several theoretical results, including: sparsity bounds for the selected coordinate set, sufficient decrease of the objective, convergence to stationary points. These results help justify the algorithmic design and provide a level of rigor that is often missing in purely heuristic acceleration methods.

4. **Strong empirical evaluation**
The experiments cover multiple datasets and tasks. Results show substantial speedups while maintaining or improving alignment accuracy.


### Weakness
1. **selection of the step size**
The selection of the step size in the current work appears to be largely empirical, and we notice that the authors adopt a **constant step size** in their algorithm. For each subproblem (i.e., Equation (5)), previous work such as [1] has shown that when a **fixed step size** is used, the algorithm’s performance may deteriorate as the problem size $n$ increases. One possible remedy is to set
$$
\beta = \gamma \ln(n)
$$
where $\gamma$ typically depends only on the problem type. In addition, [2] proposes a customized or adaptive step-size strategy, which may also serve as a useful alternative for improving the algorithm.
2. **Unclear complexity**
The complexity analysis remains somewhat unclear. Although the authors claim a complexity of $O(r^3)$, the bounds on $r$ are not well characterized. According to the figure in the paper 1, the sparsity of the coupling matrix increases as the iterations proceed. This suggests that most of the computational cost may be concentrated in the early stages of the optimization. However, the current figure does not show the sparsity behavior during the **first 50 iterations**, making it difficult to understand the computational dynamics in the initial phase.
To make the complexity analysis clearer, the authors may consider the following:
(1). Using a **log-scale plot** to show the relationship between runtime and problem size, which would allow readers to better observe how the computational complexity scales.
(2). Providing the evolution of **the sparsity behavior during the first 50 iterations**, which could help clarify how the sparsity pattern develops in the early stage and how it affects the practical computational cost.


3. **Potential overclaim.** The paper claims that the proposed method achieves a **160× speedup** compared to existing approaches. However, based on Figure 4(a), this improvement appears to be measured only relative to **the slowest baseline method** at the **2000-node scale**, rather than demonstrating a 160× improvement over all baselines. Therefore, the current wording may be somewhat overstated, and the authors may consider clarifying this claim to better reflect the empirical results. In addition, the set of baselines used in the experiments does not appear to fully cover the **state-of-the-art methods for graph alignment**. For example, methods such as [1], [2], and [3] are widely regarded as strong approaches in this area. In particular, [1] and [2] optimize almost the **same objective function as the one used in this paper and are also based on the Sinkhorn framework**. Including comparisons with at least one methods of them would provide a more comprehensive evaluation of the proposed approach.

4. **Limited Evaluation**. The experimental evaluation is rather limited. We test  the algorithm on a Facebook network dataset with about 4000 nodes from [2], while the reported accuracy is only 8% under 5% noise, raising concerns about the robustness of the method. It would strengthen this paper to include experiments on more diverse datasets and provide comparisons with stronger baselines. The dataset is available at: https://github.com/BinruiShen/Adaptive-Softassign-Matching.

[1] Binrui Shen, Qiang Niu, and Shengxin Zhu. CSGO: Constrained-Softassign Gradient Optimization For Large Graph
Matching. Pattern Recognition, 2026

[2] Binrui Shen, Qiang Niu, and Shengxin Zhu. Adaptive Softassign via Hadamard-Equipped Sinkhorn. In Proceedings of the IEEE/CVF Conference on Computer Vision and Pattern Recognition, pages 17638–17647, 2024.

[3] Binrui Shen, Yuan Liang, and Shengxin Zhu. FRAM: Frobenius-regularized assignment matching with mixed-precision
computing. In The Thirty-ninth Annual Conference on Neural Information Processing Systems, 2025.

---

> ### Author Rebuttal · Authors · 2026-03-31
>
> We thank the reviewer for the insightful comments and address the concerns below.
>
> **W1:** selection of the step size
>
> In the original submission, the constant step size $\tau$ is chosen based on the empirical robustness shown **in Section 4.2 and Fig. 3(b)**.  Following this suggestion, **we evaluate the scale-dependent rule $\tau = \gamma \ln(n)$ proposed in [1], with the hyperparameter $\gamma$ selected from $\{10, 20\}$**. LoBCD-GW achieves 99.71 / 0.40 on Douban, 94.99 / 0.14 on Reddit, 100.00 / 0.26 on Synthetic, 97.20 / 0.33 on ACM, and 96.47 / 0.27 on DBLP, where each pair denotes alignment accuracy (%) / average runtime (seconds). A discussion of these scale-dependent and adaptive strategies [1, 2] will be included in the revised manuscript.
>
> **W2:** Unclear complexity
>
> To clarify the complexity analysis, **the revised Figure 1 will include the sparsity evolution over the first 50 iterations** (anonymous link: https://anonymous.4open.science/r/r-ED77/Fig1.png),  and we also provide **a log-scale plot of runtime versus problem size based on Figure 4(a)** (anonymous link: https://anonymous.4open.science/r/r-ED77/Fig2.png ). The flatter slope of LoBCD-GW in the log-scale plot empirically supports its improved complexity scaling relative to the baselines.
>
> **W3:** Potential overclaim
>
> We agree that the original “160× speedup” wording is too strong. In the revised manuscript, we will revise the Abstract and Section 4.3 to state that **LoBCD-GW achieves up to a 5.8× speedup over the second-fastest baseline (SpecGW) at the 5000-node scale**. We will also update the annotation in Figure 4(a) for clarity (anonymous link:https://anonymous.4open.science/r/r-ED77/Fig3.png).
>
>  **We additionally evaluate CSGO [1], ASM [2], and FRAM [3] on all five datasets**, and the detailed results are available at the anonymous link:https://anonymous.4open.science/r/r-ED77/Table%202.png. Among these baselines, CSGO [1] achieves the shortest runtime on Douban, and FRAM [3] achieves the shortest runtime on Reddit. LoBCD-GW achieves the highest alignment accuracy on all five datasets, and also maintains the shortest runtime on the larger datasets Synthetic, ACM, and DBLP. We will include these baselines in the revised manuscript. We also note the effective step-size strategies used in CSGO [1] and ASM [2], which may provide useful directions for improving our method.
>
> **W4:** Limited Evaluation
>
> We thank the reviewer for the constructive suggestion regarding the Facebook dataset. The previously low accuracy (8% at 5% noise) is **due to an input normalization issue** rather than a limitation of the optimization framework. In our original code, the input uses symmetrically normalized adjacency matrices. For the Facebook network, which is extremely sparse (density 1.14%), this normalization over-compresses the dynamic range and weakens the GW gradient. **Following the input treatment in [2], we replace symmetric normalization with max-value scaling for the input matrices. With this change, the accuracy at 5% noise increases from 8% to 100%. To further assess robustness, we also test 15% and 25% noise, where the accuracy remains 100%.**
>
>  Since we cannot revise the supplementary material, we provide the implementation change and running commands here. Specifically, lines 228–229 of `LoBCD_GW.py` are replaced by
>
> a_max, b_max = A.max().clamp_min(1.0), B.max().clamp_min(1.0)
> A_n, B_n = A / a_max, B / b_max
>
> and the three `tau` values in `LoBCD_GW.py` are set to 2.7. The corresponding commands are:
>
> python run_LoBCD_GW.py --dataset facebook5/facebook15/facebook25   --rho 5.1 --rho_min 0.24 --rho_decay 0.95 --eps 2.93 --sinkhorn_iters 45 --max_iter 300
>
>  **Q1:** The block selection rule is defined...at least in the first iteration?
>
> The observation is correct: **a full gradient and a full KL projection are needed only at the first iteration to initialize** $\mathcal{A}_k$. In subsequent iterations, gradient evaluation, block selection, and projection are all restricted to $\mathcal{A}_k$. This remains accurate and efficient because Lemma 3.5 guarantees that local projection preserves global feasibility, Table 6 shows that the local gradient closely matches the global gradient (error $10^{-3}$ to $10^{-5}$, cosine similarity 0.99-1.00), and we verify that recomputing the full gradient every 50 iterations causes almost no change in alignment accuracy.
>
> **Q2:** Could the authors provide...across different problem sizes.
>
> The runtimes (in seconds) of LoBCD-GW at each node scale in Figure 4 are listed below.
>
> | Number of nodes | Running Time(seconds) |
> | :-------------- | :-------------------- |
> | 1000            | 0.4658                |
> | 2000            | 1.8551                |
> | 3000            | 1.9682                |
> | 4000            | 3.6038                |
> | 5000            | 5.2929                |

---

> > ### Author Rebuttal · Reviewer_V5Wi · 2026-04-03
> >
> > Thank you very much to the reviewer for the detailed response and the effort invested, which has addressed most of my concerns. I would like to further ask:
> >
> > In the comparison with CSGO, ASM, and FRAM, does LoBCD-GW utilize a GPU while the other three algorithms are implemented on a CPU?
> >
> > The authors acknowledged that the first iteration requires the entire gradient matrix. Does this imply that the computational complexity remains O(n^3)?

---

> > > ### Author Response · Authors · 2026-04-07
> > >
> > > We thank the reviewer again for the thoughtful comments and address the additional concerns below.
> > >
> > > **Q1:** In the comparison with...are implemented on a CPU?
> > >
> > > Yes, those three baselines are run on CPU following their original code settings. We thank the reviewer for highlighting this fairness concern.
> > >
> > > To further address your concern, **we additionally run LoBCD-GW on CPU only and report the CPU runtimes** for a fair CPU-to-CPU comparison. The CPU results are obtained by running the same PyTorch-based implementation with the same hyperparameter settings. These experiments are conducted on the same server equipped with an Intel Xeon Platinum 8358P CPU. On the five datasets Douban, Reddit, Synthetic, ACM, and DBLP, **the corresponding CPU runtimes are 0.52, 0.57, 1.14, 0.63, and 0.65 seconds.** (If using GPU, our runtimes are 0.45, 0.22, 0.27, 0.49, and 0.52 seconds, respectively). We further **compare the CPU runtimes of the three baselines with the newly measured CPU runtimes of LoBCD-GW**. The detailed results are available at the anonymous link: https://anonymous.4open.science/r/r-ED77/Table%202%20(new).png. CSGO is the fastest on Douban (0.10 s), reflecting the effectiveness of its step-size and warm-start strategy. FRAM is the fastest on Reddit (0.03 s), and **LoBCD-GW is the fastest on the larger datasets Synthetic, ACM, and DBLP**, which demonstrates the effectiveness of our localized block coordinate descent strategy. We will incorporate these results and the corresponding hardware annotations into the revised manuscript.
> > >
> > > **Q2:** The authors acknowledged...the computational complexity remains ${\mathrm{O}}(n^{3})$?
> > >
> > > We thank the reviewer for the concern about the computational complexity of initializing $\mathcal{A}_k$. The first iteration indeed **incurs a one-time** $\boldsymbol{\mathrm{O}}(n^{3})$ full-gradient computation, and we will make this explicit in the revised manuscript. **However, this does not imply $\boldsymbol{\mathrm{O}}(n^{3})$ complexity at every iteration.** After initialization, LoBCD-GW restricts gradient evaluation to the selected set $\mathcal{A}_k$, **yielding a per-iteration complexity of** $\boldsymbol{\mathrm{O}}(r^{3})$ with $r \ll n$. Accordingly, under the same stopping criterion, the total runtime can be more accurately characterized as $\boldsymbol{\mathrm{O}}(n^{3} + (K-1)r^{3})$, where $K$ is the number of iterations. Specifically, in our GPU implementation, **the first full-gradient step accounts for only 1.54%, 3.03%, 3.49%, 7.07%, and 6.95% of the total runtime on Douban, Reddit, Synthetic, ACM, and DBLP, respectively; in our CPU implementation, the corresponding percentages are 0.96%, 1.14%, 4.28%, 6.60%, 3.52%.** In other words, **the initialization cost accounts for a small portion of the total runtime**. Moreover, the sparsity evolution (anonymous link: https://anonymous.4open.science/r/r-ED77/Fig1.png) shows that the selected set $\mathcal{A}_k$ shrinks rapidly during optimization, and Section 4.3 (Scalability Analysis) further reveals that **this one-time initialization cost can be significantly amortized in practice**. Finally, regarding the initialization cost, we think a potential improvement could be using the warm-start technique [1] or some other approximation ideas. We will add more discussions to the revised manuscript.
> > >
> > > [1] Binrui Shen, Qiang Niu, and Shengxin Zhu. CSGO: Constrained-Softassign Gradient Optimization For Large Graph Matching. Pattern Recognition, 2026

---

### Official Review · Reviewer_w3kk · 2026-03-10

**Soundness:** 2
**Presentation:** 3
**Significance:** 2
**Originality:** 3
**Overall Recommendation:** 4
**Confidence:** 3

**Summary:**

The Gromov-Wasserstein (GW) distance provides a powerful framework for aligning structured data by comparing the intrinsic geometries of metric measure spaces, and has become a fundamental tool in machine learning. However, its high time complexity remains a major bottleneck in large-scale applications, severely limiting the scalability. To address this challenge, this paper propose LoBCD-GW, an efficient GW optimization algorithm. Specifically, they reveal the data-dependent sparsity of large-magnitude updates to the coupling matrix and introduce a localized block coordinate selection strategy. This confines the optimization to a “selected set” of size $r$ thereby reducing the complexity to $O(r^3)$. They conducted a set of experiments on various datasets, and the results demonstrate that their method is superior to baselines.

**Compliance With Llm Reviewing Policy:**

Affirmed.

**Final Justification:**

The authors clarify the theoretical issues I raised and provide more supplemental experiments. New results enhances the effectiveness of their proposed method.

**Key Questions For Authors:**

Please see the questions in weaknesses.

**Limitations:**

yes

**Strengths And Weaknesses:**

Strengths:

(1) They confine the computationally intensive updates to a selected set which is identified by large constrained gradient magnitudes, reducing the per-iteration complexity from $O(n^3)$ to $O(r^3)$;

(2) Unlike constraint relaxation methods, they propose a novel “marginal compensation mechanism” for localized block coordinate descent that guarantees strict feasibility;

(3) They proved the convergence of our algorithm and demonstrate superior efficiency and state-of-the-art accuracy on large-scale graph benchmarks.

Weaknesses:

(1) There exist several methods for computing GW distance. For example, [1] proposed a proximal point algorithm (PPA) for solving the GW problem. [2] introduced the Sampled GW. [3] developed Spar-GW, which approximates the GW distance via a sparse coupling matrix. more recently, [4] proposed FGC-GW, which reduces the computational complexity from cubic to quadratic. Could the authors include these as their baselines and report the numerical results? It would strengthen the empirical evaluation if the authors could include these methods as additional baselines.

[1] Xu, H., Luo, D., Zha, H., & Duke, L. C. (2019, May). Gromov-wasserstein learning for graph matching and node embedding. In *International conference on machine learning* (pp. 6932-6941). PMLR.

[2] Kerdoncuff, T., Emonet, R., & Sebban, M. (2021). Sampled gromov wasserstein. *Machine Learning*, *110*(8), 2151-2186.

[3] Li, M., Yu, J., Xu, H., & Meng, C. (2023). Efficient approximation of Gromov-Wasserstein distance using importance sparsification. *Journal of Computational and Graphical Statistics*, *32*(4), 1512-1523.

[4] Zhang, W., Wang, Z., Fan, J., Wu, H., & Zhang, Y. (2024). Fast gradient computation for Gromov-Wasserstein distance. *arXiv preprint arXiv:2404.08970*.

(2) The paper evaluates the proposed method and baselines using graph alignment accuracy. It would be helpful if the authors could also report numerical results comparing the GW distance obtained by LoBCD-GW with the GW distance computed using a standard solver, such as the implementation provided in the POT package. This comparison could help assess how closely LoBCD-GW approximates the true GW distance.

(3) Lines 128-130 state 'Once the optimal coupling $\pi^*$ is obtained, let $\mathcal{M}$ dempte the set of all feasible point-to-point (injective) mappings ...' The GW distance corresponds to a constrained non-convex minization problem, for which the global optimum cannot generally be obtained via gradient descent. Additionally, Could the author further clarify how a feasible point-to-point mapping is derived from a coupling $\pi$?

(4) Equation (3) appears to follow directly from the formulation in Peyre et al. 2016. Since this result is already well established in prior work, this section could potentially be simplified, and it seems unnecessary to give the proof in this paper again.

(5) The implication of Theorem 3.8 is somewhat unclear. Could the authors clarify what conditions a stationary point of the original GW problem follows? In particular, does $\|G_\tau(\pi)\|_F=0$ imply that $\nabla L(\pi)=0$?

(6) It would strengthen the experimental results if the authors reported performance using mean plus standard deviation over multiple runs. This would provide a more reliable comparison between methods.

(7) In Table 7, LoBCD appears to perform worse than several non-GW methods on both the Amazon and Village datasets. Could the authors provide further discussion or analysis explaining this behavior?

(8) In Table 9, are there numerical results for the tasks $\text{D}\rightarrow\text{Ch}$ and $\text{Ch}\rightarrow\text{D}$? If so, it would be helpful to include them in the table for completeness.

---

> ### Author Rebuttal · Authors · 2026-03-31
>
> We thank the reviewer for the thoughtful comments, and address your concerns below.
>
> **W1:** There exist several methods...as additional baselines.
>
> First, we clarify that the **proximal-point-based methods in [1] are already included** in our original manuscript as **BPG and BPG-S (see Table 1)**. We further evaluate the other three methods, **Sampled-GW [2], SparGW [3], and FGC-GW [4]**, and report their results at the anonymous link https://anonymous.4open.science/r/r-ED77/Table1.png
>
> The results show that LoBCD-GW consistently achieves the shortest runtime, while SparGW obtains the highest accuracy on Douban. We will add these baselines to the revised manuscript.
>
> **W2:** The paper evaluates...approximates the true GW distance.
>
> We additionally **compare LoBCD-GW with the standard FW solver in the POT package by reporting the final GW objective values**. Due to space limitations, the dataset construction and full loss table are provided in our response to Reviewer pEtp (W1/W2/Q1/Q4). On the Gaussian point cloud dataset, LoBCD-GW achieves a GW loss of $2.96 \times 10^{-5}$, close to the zero loss attained by FW. On the SBM-generated graphs, LoBCD-GW attains a lower GW loss than FW. **This suggests that LoBCD-GW closely approximates the standard solver on the small continuous instance and may be less affected by poor local minima on the SBM-generated graphs.**
>
> **W3:** Lines 128–130 state ... derived from a coupling $\pi$?
>
> We agree that “optimal” is too strong for the non-convex GW problem.  **We will revise “optimal coupling” to “optimized coupling.”**
>
> We will also clarify that the feasible point-to-point mapping is obtained in a **post-processing step** from the coupling $\pi$, rather than from the GW optimization itself. Concretely, as defined in Eq. (2), the discrete mapping $M^*$ is extracted from the soft coupling by solving a maximum-weight injective matching problem, implemented exactly by **the Hungarian algorithm or approximately by a greedy row-wise "argmax" on very large graphs.**
>
> **W4:** Equation (3) ...in this paper again.
>
> We agree. In the revised manuscript, we will remove the proof from the appendix and retain Eq. (3) only to introduce $L(\pi)$ and its gradient for later use.
>
> **W5:** The implication of...imply that $\nabla L(\pi) = 0$?
>
> **To clarify, $\|G_{\tau}(\pi)\|_F = 0$ does not imply the unconstrained condition $\nabla L(\pi) = 0$.** Since the GW problem is constrained to the transport polytope $\Pi(\mu, \nu)$, the stationary point in Theorem 3.8 is a **first-order KKT stationary point of the constrained problem**.
>
> Specifically, Eq. (30) shows that $\|G_{\tau}(\pi)\|_F = 0$ means that $\pi$ is a fixed point of the KL-proximal update. In this case, **the reduced gradient $G_{ij}$ (Equations 16-18) satisfies the KKT conditions**: it is zero on the support of $\pi$ and nonnegative elsewhere. **Therefore, Theorem 3.8 establishes stationarity in the constrained KKT sense, rather than the unconstrained condition $\nabla L(\pi) = 0$.**
>
> **W6:** It would strengthen...between methods.
>
> For completeness, we repeated LoBCD-GW over 5 runs and report mean ± standard deviation. **Since both the algorithm and its initialization are deterministic, the accuracy is unchanged across runs and is therefore omitted here. Only runtime varies slightly: Douban 0.45±0.06, Reddit 0.22±0.13, Synthetic 0.27±0.09, ACM 0.49±0.07, and DBLP 0.52±0.11.** **Table 4** of the original manuscript already provides 5-run statistics under noisy settings with different random seeds.
>
> **W7:** In Table 7...or analysis explaining this behavior?
>
> We believe this gap mainly stems from limitations of the GW formulation for partitioning tasks.
>  **(1) Idealized topology:** Modeling partitioning as alignment to a disconnected prototype (the identity matrix) ignores the inter-community edges prevalent in complex networks such as Amazon and Village.
> **(2) Constraint mismatch:** GW enforces strict mass conservation, which may conflict with highly imbalanced, power-law community sizes and thus cause unnatural splitting or merging. In contrast, non-GW methods directly optimize modularity and better fit such topologies.
>
> **W8:** In Table 9...include them in the table for completeness.
>
> We will add the results for D $\leftrightarrow$ Ch to Table 9 in the revised manuscript for completeness.
>
> | Method   | D→Ch     | Ch→D     | Time (s) |
> | -------- | -------- | -------- | -------- |
> | eBPG     | 54.2     | 38.6     | 92.38    |
> | BPG-S    | 25.2     | 25.6     | 24.98    |
> | BPG      | 25.0     | 25.6     | 40.50    |
> | specBPG  | 38.2     | 39.3     | 8.02     |
> | FW       | 38.5     | 39.3     | 36.39    |
> | LR       | 60.3     | 58.4     | 16.29    |
> | KL-BAPG  | 39.4     | 34.5     | 1.78     |
> | ScalaGW  | **82.2** | 38.6     | 40.10    |
> | LoBCD-GW | 73.5     | **42.5** | **0.69** |

---

> > ### Author Rebuttal · Reviewer_w3kk · 2026-04-02
> >
> > Thanks the authors for their response. The supplemental experiments have reduced my concern. I increase my rating in response to the improvement.

---

> > > ### Author Response · Authors · 2026-04-02
> > >
> > > We thank the reviewer again for the thoughtful comments and valuable suggestions.

---

### Official Review · Reviewer_k15m · 2026-03-15

**Soundness:** 2
**Presentation:** 2
**Significance:** 3
**Originality:** 3
**Overall Recommendation:** 4
**Confidence:** 4

**Summary:**

This paper proposes LoBCD-GW, an algorithm for computing the Gromov-Wasserstein (GW) distance that exploits the data-dependent sparsity of coupling matrix updates. The key observation is that most coupling entries stabilize quickly during optimization, so significant gradient updates are confined to a shrinking subset of sample pairs. The authors introduce a localized block coordinate selection strategy that restricts updates to a "selected set" of size $r \ll n$, reducing per-iteration complexity from $O(n^3)$ to $O(r^3)$. The method uses mirror descent with KL divergence regularization. Convergence is proved and experiments on graph alignment benchmarks show a speedup over existing methods while maintaining accuracy.

**Compliance With Llm Reviewing Policy:**

Affirmed.

**Final Justification:**

The detailed answers (from the rebuttal) should be integrated into the final paper.

**Key Questions For Authors:**

1. **Frank-Wolfe comparison:** How does LoBCD-GW compare to a standard Frank-Wolfe algorithm on the GW objective (see https://github.com/jovo/FastApproximateQAP)? A direct comparison would clarify the algorithmic contribution.

2. **Step size $\tau$:** How is the step size $\tau$ selected? Is there a principled rule (e.g., line search, schedule), or is it tuned per dataset? How sensitive are the results to this choice?

3. **JOENA discrepancy:** Why do the JOENA results in Table 1 differ from those in the original paper? Have you verified your JOENA implementation against the original code and settings? Running LoBCD-GW on the original JOENA benchmarks would resolve this concern.

**Limitations:**

The authors do not discuss the sensitivity of the method to the step size $\tau$ or the selection threshold $\epsilon_\Delta$. The fairness of timing comparisons across CPU/GPU implementations is not addressed.

**Strengths And Weaknesses:**

- **Originality:** The data-dependent sparsity observation and localized block coordinate descent strategy for GW optimization is a well-motivated algorithmic contribution.
- **Significance:** The speedups are substantial and address a genuine computational bottleneck in GW computation.
- **Soundness -- comparison with Frank-Wolfe:** The algorithm is based on mirror descent with a KL divergence term (Equation 5). The authors should compare their method with a standard Frank-Wolfe algorithm applied to the same GW objective to isolate the contribution of the KL regularization. What is the specific impact of the KL term on convergence and solution quality?
- **Significance -- missing baseline (graph alignment):** For the graph alignment experiments, the authors should compare against Fast Approximate QAP (FAQ), which is a well-established baseline for this task. Its absence weakens the experimental evaluation.
- **Presentation -- step size selection:** The step size $\tau$ plays a central role in the algorithm (Equations 5-6) but its selection is not discussed. How is $\tau$ chosen in practice? Is the method sensitive to this choice?
- **Soundness -- timing comparisons (Table 1):** It is unclear how the authors handle the comparison of running times across different algorithms. Some methods may use GPU acceleration while others run on CPU. The authors should clarify the hardware setup for each algorithm and ensure fair timing comparisons.
- **Soundness -- discrepancy with JOENA results:** The results reported in Table 1 for JOENA differ from those in the original JOENA paper. The authors should use the same experimental setting as the original paper and verify that their JOENA implementation is correct. Alternatively, they should test LoBCD-GW on the exact benchmarks from the JOENA paper to ensure a fair comparison.

---

> ### Author Rebuttal · Authors · 2026-03-31
>
> We thank the reviewer for the thoughtful comments, and address your concerns below.
>
> **W1/Q1:** Soundness -- comparison with Frank-Wolfe
>
> **We clarify that the FW baseline in Table 1 is the standard Frank-Wolfe algorithm applied to the same GW objective in Eq. (3)**, so Table 1 already provides a direct comparison under the same objective. **LoBCD-GW achieves higher alignment accuracy than FW. To further compare solution quality beyond alignment accuracy, we additionally report GW objective values on two new datasets.** Due to space limitations, the dataset construction and full loss table are provided in our response to Reviewer pEtp (W1/W2/Q1/Q4). On the Gaussian point cloud dataset, LoBCD-GW achieves a GW loss of $2.96 \times 10^{-5}$, close to the zero loss attained by FW. On the SBM-generated graph dataset, LoBCD-GW achieves a lower GW loss than FW (0.033 vs. 0.054). These results suggest that LoBCD-GW is close to the standard solver on a continuous dataset and attains a better objective value on a dataset more prone to poor local minima.
>
> Regarding the KL term, its main role is on the optimization side: it induces the KL-proximal / mirror-descent update in Eqs. (5)–(6), which keeps each iterate feasible after projection onto $\Pi(\mu, \nu)$. **The KL-proximal update is the update form for which Theorem 3.8 establishes convergence to a stationary point.** The comparison with FW supports the advantage of the overall LoBCD-GW optimization scheme under the same GW objective, rather than isolating the effect of the KL term alone. In particular, on the SBM dataset, the lower GW loss is more appropriately attributed to the combined effect of the KL update and localized block selection, which helps focus optimization on the most critical coupling entries while preserving feasibility and may reduce the risk of poor local minima in practice.
>
> **W2/Q1:** missing baseline
>
> **We will add FAQ, a Frank-Wolfe-type graph-matching baseline, in the revised manuscript.**
>
> For the FAQ baseline, the alignment accuracy (%) / average alignment time per graph pair (seconds) are **85.33 / 1.81 on Douban, 40.20 / 4.99 on Reddit, 64.62 / 42.96 on Synthetic, 92.53 / 12.27 on ACM, and 79.83 / 20.99 on DBLP.**
>
> **W3/Q2:** Presentation -- step size selection
>
>  In practice, $\tau$ is treated as a tunable hyperparameter. **As stated in Section 4.1 (Parameter Setup), it is selected by validation from $\{2, 5, 10, 20, 100\}$.** **Our method is not highly sensitive to the choice of the step size $\tau$. As shown in Section 4.2 (Parameter Sensitivity Analysis) and Figure 3(b)**, LoBCD-GW remains stable over a broad range of $\tau$, and **$\tau \in [10, 100]$ yields stable performance** across the five benchmark datasets.
>
> **W4:** Soundness -- timing comparisons
>
> We thank the reviewer for concern about runtime.  All experiments are conducted **on the same server equipped with an Intel Xeon Platinum 8358P CPU and an NVIDIA A100-SXM4 GPU**. For fairness, each baseline is run in its natural implementation setting. Specifically, **FW, SpecGW, ScalaGW, and FAQ run on CPU**, since their implementations rely on LP-based inner solvers, recursive graph procedures, or other operations whose available implementations are CPU-based rather than GPU-oriented. **LoBCD-GW, eBPG, BPG, BPG-S, KL-BAPG, JOENA, LR, and HOT run on GPU**, as their core computations are dominated by Sinkhorn iterations, tensor operations, or low-rank matrix multiplications, which are well suited to GPU acceleration. We will clarify this hardware setup explicitly in the revised manuscript.
>
> **W5/Q3:** Soundness -- JOENA discrepancy
>
> We verified that the implementation follows the original code. **The difference between the JOENA results in Table 1 and those reported in the JOENA paper is due to the benchmarks being different.** As detailed in Section 4.1 (Datasets) and Appendix Table 2, our datasets (sourced from HOT, Zeng et al., 2024) are Douban (avg. 234 nodes, 345.33 edges), ACM (avg. 1000 nodes, 4163 edges), and DBLP (avg. 1000 nodes, 3747 edges). We chose these because they provide multiple graph pairs, enabling statistically averaged results across diverse topologies. JOENA evaluates on single graph pairs (ACM: 9,872 nodes, 39,561 edges; DBLP: 9,916 nodes, 44,808 edges; Douban: 3,906 nodes, 16,328 edges). To eliminate this naming ambiguity, we will rename our datasets to ACM-1000, DBLP-1000, and Douban-230.
>
> **Response to Limitations**
>
> We discuss the **sensitivity of LoBCD-GW to the step size $\tau$ and the selection threshold $\epsilon_{\Delta}$ in Section 4.2 (Parameter Sensitivity Analysis)** of the original manuscript. As shown in Figure 3, the experimental results demonstrate that our method is **insensitive** to these two parameters.  Regarding timing fairness, each baseline is run in its natural implementation setting, and **we will explicitly clarify the corresponding CPU/GPU hardware setup.**

---

> > ### Author Rebuttal · Reviewer_k15m · 2026-04-03
> >
> > Thank you for the detailed answers, which should be integrated into the final paper. I increase my rating in response to the improvement.

---

> > > ### Author Response · Authors · 2026-04-03
> > >
> > > Thank you again for your valuable comments and feedback.

---

### Decision · Program_Chairs · 2026-04-30

**Decision:**

Accept (regular)

**Comment:**

This paper studies scalable optimization for Gromov-Wasserstein distance and introduces LoBCD-GW, a localized block coordinate descent method that exploits the empirical sparsity of meaningful coupling updates by focusing computation on a dynamically selected active set together with a marginal compensation mechanism that preserves feasibility. The main contributions are this localized active-set algorithm, theoretical analysis of sparsity, descent, and convergence to constrained stationary points, and strong empirical results on graph alignment and related tasks showing substantial speedups with little loss in solution quality. The main strengths are the originality and practical relevance of the algorithmic idea, the fact that it is more than a simple heuristic thanks to the compensation mechanism and supporting theory, and a strong empirical story that became more convincing after rebuttal. The main weaknesses concern the original overstatement of the complexity and speedup claims, the fairness and completeness of some experimental comparisons, and several optimization-detail clarifications; the rebuttal addressed these issues well by correcting the complexity framing, adding fairer CPU-only and stronger baseline comparisons, and clarifying the optimization interpretation, although the strongest speedup claims should still be toned down in the final version. Overall, the paper makes a solid and practically useful contribution to scalable GW optimization, and I recommend acceptance.